

# Quantifying organic matter and functional groups in particulate matter filter samples from the southeastern United States, part I: Methods

Alexandra J. Boris[1], Satoshi Takahama[2], Andrew T. Weakley[1], Bruno M. Debus[1], Carley D. Fredrickson[3], Martin Esparza-Sanchez[1], Charlotte Burki[2], Matteo Reggente[2], Stephanie L. Shaw[4], Eric S. Edgerton[5], Ann M. Dillner[1]

[1]Air Quality Research Center, University of California Davis, Davis, CA, United States
[2]ENAC/IIE Swiss Federal Institute of Technology Lausanne (EFPL), Lausanne, Switzerland
[3]Atmospheric Science Department, University of Washington, Seattle, WA, United States
[4]Electric Power Research Institute, Palo Alto, CA, 94304, United States
[5]Atmospheric Research & Analysis, Inc., Cary, NC 27513, United States

*Correspondence to:* Ann M. Dillner (amdillner@ucdavis.edu)

**Abstract.** Comprehensive techniques to describe the organic composition of atmospheric aerosol are needed to elucidate pollution sources, gain insights into atmospheric chemistry and evaluate changes in air quality. Fourier Transform Infrared absorption (FT-IR) spectrometry can be used to characterize atmospheric organic matter (OM) and its composition via functional groups on aerosol filter samples in air monitoring networks and research campaigns. We have built FT-IR spectrometry functional group calibration models that improve upon previous work. Laboratory standards that simulated the breadth of the absorbing functional groups in atmospheric OM were made: particles of relevant chemicals were first generated, collected, and analyzed. Challenges of collecting atmospherically relevant particles and spectra were addressed by including interferences of particle water and other inorganic aerosol constituents and exploring the spectral effects of inter-molecular interactions. Calibration models of functional groups were then constructed using partial least squares (PLS) regression and the collected laboratory standard data. These models were used to quantify concentrations of five organic functional groups and OM in eight years of ambient aerosol samples from the southeastern aerosol research and characterization (SEARCH) network. The results agreed with values estimated using other methods, including thermal optical reflectance (TOR) organic carbon (OC; $R^2=0.74$) and OM calculated as a difference between total aerosol mass and inorganic species concentrations ($R^2=0.82$). Comparisons with previous calibration models of the same type demonstrate that this new, more complete suite of chemicals has improved our ability to estimate oxygenated functional group and overall OM concentrations. Calculated characteristic and elemental ratios including OM/OC, O/C and H/C agree with those from previous work in the southeastern US, substantiating the aerosol composition described by FT-IR calibration. The median OM/OC ratio over all sites and years was 2.1±0.2. Further results discussing temporal and spatial trends of functional group composition within the SEARCH network will be published in a forthcoming article.



# 1    Introduction

## 1.1    Challenges of quantifying atmospheric aerosol organic matter mass

Atmospheric aerosol organic matter (OM) composition, sources, and formation processes have been a focus of research for many decades (Haagen-Smit, 1952; Went, 1960). However, because the organic fraction of aerosol particles contains

thousands of individual chemical compounds (Schum et al., 2018), it is a difficult task to characterize the total OM composition of an aerosol sample. Typical molecular-level analytical techniques quantify up to 30% of OM concentration (Hallar et al., 2013). Chromatography techniques suffer from the need to have sufficient molecular selectivity and sensitivity for each chemical, requiring calibration of each species. As an alternative, rather than quantifying each chemical, the total OM concentration can be measured.

Some analytical techniques such as aerosol mass spectrometry can quantify OM concentrations in real time (Aiken et al., 2008). Other methods with involved chemical analyses of discrete filter samples have been used to estimate OM concentration. These include mass balance of aerosol constituents using various analytical techniques (El-Zanan et al., 2009; Simon et al., 2011; Xing et al., 2013), chemical mass balance and molecular markers measured by chromatography and mass spectrometry (Chen et al., 2012), extrapolation from gas chromatography/mass spectrometry of extracts (Turpin and Lim,

2001), infrared absorption spectrometry of extracts (Polidori et al., 2008), or thermal–optical and gravimetric analyses of extracts (El-Zanan et al., 2009). However, each of these methods is subject to specific limitations. Aerosol mass spectrometry OM concentrations, for example, are subject to uncertainties resulting from fragmentation and high heat exposure (Canagaratna et al., 2015). Filter extraction procedures can result in the loss of organic species (Kawamura and Bikkina, 2016) and render a sample unusable for further analysis, while mass balance techniques require accurate estimation

of all non-organic species concentrations, which can involve large uncertainties (e.g., particle water and losses of volatile nitrate during $NH_4NO_3$ collection onto Nylon filters; Chow et al., 2015).

The methods of estimating OM concentrations listed above are either not feasible or have considerable uncertainty for measurements that are remote, resource-limited, or long-term (e.g., multi-year). Analyses of OM for routine monitoring networks have specific requirements. Because of the large number of samples, collection must be simple, and the cost of

analysis must be low. Non-destructive, filter-based techniques are also desirable for networks because they allow for multiple chemical analyses to be performed on one sample.

In air monitoring networks, OM concentrations are typically estimated indirectly from organic carbon (OC) concentrations (Edgerton et al., 2005; Pitchford et al., 2007). While OM includes other atoms such as O and H associated with C (sometimes also N, S, and P; Russell, 2003), OC accounts for only the C atoms. Sample OM concentration is typically

determined from thermal optical reflectance (TOR) OC by multiplying the OC concentration by a static ratio of OM/OC. An OM/OC value of 1.4 for urban samples (White et al., 1977) or 1.8 for rural samples (Pitchford et al., 2007) is typically used. However, OM/OC varies widely amongst ambient samples. For example, Ruthenburg et al. (2014) estimated values varying



between 1.46 and 2.01 (10[th] and 90[th] percentiles) in just one year of regular OM filter samples at seven rural US locations. This and other observed OM/OC variability suggest that a static value of OM/OC is not adequate to capture the spatial and temporal variations in OM. A technique for routine OM concentration measurement in ambient aerosol at network sites is therefore needed.

## 1.2  Using infrared absorption of functional groups to quantify aerosol OM

Fourier transform infrared absorption (FT-IR) spectrometry can be used to quantify nearly the entire organic aerosol concentration in a given sample by functional groups (Coury and Dillner, 2009; Faber et al., 2017; George et al., 2015; Reff et al., 2005; Russell et al., 2011; Ruthenburg et al., 2014). Measuring functional group concentrations in ambient aerosol samples is useful to: (1) accurately estimate the total OM concentration; (2) further characterize the OM composition by functional groups; (3) monitor organic composition and sources of aerosol over time; and (4) estimate the degree of oxidation. The FT-IR spectrometry approach is particularly useful for routine/network OM measurements because it can be applied to filter samples that are routinely collected for other purposes (e.g., particulate matter mass), is non-destructive, and is inexpensive.

The principle of organic characterization through FT-IR spectrometry is as follows: chemical bonds with appropriate vibrational symmetries and frequencies absorb light at specific mid-infrared wavelength ranges, allowing the determination of the bond type and, in some cases, even molecular environment. The magnitude of the light absorption is proportional to the number of bonds present, allowing the direct quantification of bonds within an aerosol sample (Allen et al., 1994).

Infrared absorption spectrometry has been used to quantify functional groups using a peak-fitting approach (Takahama et al., 2013), but factor-based calibration of spectra can more readily determine interferents and is strengthened by using multiple spectral bands at once (Naes et al., 2002). Specifically, partial least squares (PLS) regression has been used in factor-based work. A comparison of the peak fitting and PLS calibration methods has been recently discussed elsewhere (Reggente et al., 2018). In a PLS functional group calibration, concentrations of pure chemical standards are regressed onto their corresponding FT-IR spectra to reduce the number of variables describing the data. These new variables, sometimes called "factors", are identified to explain the covariance between the chemical standard concentrations and spectra. Each functional group is quantified as a weighted sum of the extracted factors, resulting in a unique calibration model for each functional group (see Sect. 2.5, Supplementary Material Sect. 10, and Naes et al., 2002). Examples of PLS calibration of functional groups in atmospheric OM include the work of Reff et al. (2007), Coury and Dillner (2009), Ruthenburg et al. (2014), and Kamruzzaman et al. (2018).

Calibration curves are developed from "laboratory standards": pure chemicals collected onto fresh PTFE filters. The chemical mass collected is varied in mass to capture the relationship between infrared absorption and number of bonds (Coury and Dillner, 2008). Ruthenburg et al. built a set of FT-IR/PLS calibration models using nine organic chemicals and one inorganic salt interferent (ammonium sulfate) to quantify four functional groups: aliphatic C-H, carbonyl (C=O),



carboxylic acid O-H, and alcoholic O-H. The concentrations of these functional groups (and OM concentrations as weighted sums of these functional groups) were predicted in ambient filter samples from seven IMPROVE network sites collected in 2011. The same measurements were made, adding an amine functional group model, for a broader selection of IMPROVE network sites from 2013 (Kamruzzaman et al., 2018). However, the relatively short list of chemicals to represent

atmospheric composition likely limited the ability of these models to characterize the aerosol composition fully. Previous work was also done with a more comprehensive list of chemical standards; unfortunately, the particular measurement technique damaged the filter samples, which is not desirable for air monitoring network data (Coury and Dillner, 2008).

## 1.3 Functional group calibration method improvements

Efforts to improve previous FT-IR functional group concentration measurements involve addressing the following

challenges: (1) approximating atmospheric composition by selecting appropriate lists of chemicals and functional groups for calibration; (2) considering ambient aerosol molecular environments, including particle water content; (3) selecting appropriate model parameters based on the current understanding of atmospheric composition; (4) validating models when methods for direct comparison are lacking; and (5) quantifying as much of the OM as possible given that most, but not all, relevant molecular bonds absorb in the mid-infrared spectral range. The following paragraphs discuss these challenges in

more detail.

The selection of pure chemicals is non-trivial: atmospherically representative bonds must be selected to allow the calibration to capture the variation in ambient samples. It is not possible to generate standards of the thousands of individual molecules that exist within aerosol samples, many of which have not yet been characterized (Schum et al., 2018). An appropriate starting point for the list of chemical standards used in the calibration models is the atmospheric speciation reported in

previous studies. The molecular bonds, or functional groups, included in the calibration must represent the majority of the OM.  In addition, efforts to measure sub-groups of functional groups within a broad functional group category such as carbonyl groups are made (e.g., inclusion of dicarboxylic acids and amino acids), while recognizing the limitations of subdividing groups given overlapping spectral features. In addition, inorganic species that absorb infrared light must also be included as "interferents" in a robust calibration model.

Laboratory standards are prepared with the goal of capturing the molecular structures and intermolecular interactions most relevant for the atmosphere. The infrared spectrum of a molecule is affected by its chemical environment, including its hydrogen and ionic bonding interactions with other molecules in a sample (Davey et al., 2006; Mayo et al., 2003). Ideally, the variety of interactions between the many molecules in ambient aerosol particles would be modeled by the calibration to capture the variability in infrared spectral features. The bonding structures within particles of single, pure chemicals, and

between mixtures of chemicals, may also warrant consideration. Mixtures can probe for interactions between different types of polar, organic functional groups (hydrogen bonding), as well as organic with inorganic ions (ionic bonding, such as carboxylates). Water chemically or physically bound to collected ambient aerosol particles is also expected to alter ambient





samples spectroscopically and could be abundant (Dabek-Zlotorzynska et al., 2011). The presence of water could induce molecular transitions such as formation of gem-diols from carbonyls (Maroń et al., 2011), or enhance spectral features of particle water: as liquid water associated with particles (Faber et al., 2017), or as hydrate water chemically bound to particle chemical constituents (Cziczo and Abbatt, 2000). Laboratory generated particles under humid conditions may display these

spectral impacts of water, and may be useful as inputs to inform models.

The inputs to PLS models must be carefully selected to minimize measurement uncertainty. Examples of inputs includes the concentration range of the chemical standards and the number of PLS factors included in each model. These inputs are selected based on the best available information but may need to be updated over time as understanding of atmospheric composition improves.

Few methods exist for verifying FT-IR spectrometry functional group concentrations. Strong correlations have been found between ratios of FT-IR spectrometry measurements with high-resolution aerosol mass spectrometry tracer ions (e.g., ratioed carboxylic acids and C-H groupings); direct (not ratioed) correlations between measurements were less successful (Faber et al., 2017; Russell et al., 2009). Ruthenburg and colleagues (2014) quantitatively evaluated their FT-IR functional group concentrations by comparing OC concentrations from summed functional groups with TOR OC concentrations.

Although comprehensive in that a broad range of molecules in OM are detected, there are some limitations to the sensitivity of FT-IR spectrometry. Some bonds such as tertiary C-C bonds and C-O bonds do not absorb in mid-infrared spectral regions, or absorb where the filter substrate, polytetrafluoroethylene (PTFE), also absorbs (Weakley et al., 2016). Ongoing work using empirically based simulations aims to quantify this "mass recovery" of FT-IR spectrometry resolvable ambient OM (forthcoming work by Burki et al.).

**1.4  Summary of study goals**

The goal of this work is to further develop a method to measure functional group concentrations and calculate OM concentrations in ambient aerosol samples using FT-IR spectrometry and PLS calibration. Samples were collected by the southeastern aerosol research and characterization (SEARCH; Hansen et al., 2003) network. There are two main components of achieving the overall study goal. The first is to expand upon previous work (Ruthenburg et al., 2014) to better characterize

OM and address other challenges of FT-IR calibration (as described in Sect. 1.3). The second is to evaluate the improved method by quantifying atmospheric functional group concentrations over a large filter dataset of multiple years at consistent locations.

To address the first component of achieving the study goal, a broader list of atmospherically relevant chemical standards were included, such as chemicals specific to the southeastern US. The functional groups included more specific subgroups

than in previous work: aliphatic C-H groups, carboxylic acids, oxalates, non-oxalate and non-acid carbonyls, and alcohols. Additional interfering species, including particle water and ammonium nitrate, were accounted for, and molecular



interactions expected in ambient samples were considered. Model parameters such as the number of regression factors were selected based in part on current atmospheric composition literature and focused studies using simulation methods.

To address the second component, SEARCH samples from 2009–2016 at five sampling sites with varying (urban/rural) emissions were analyzed. The calibration of SEARCH samples was particularly challenging due to interference from the

thicker filter material and lower aerial density of particles than the IMPROVE samples used by Ruthenburg et al., 2014. The final models were evaluated qualitatively and semi-quantitatively by comparing the ambient SEARCH functional group measurements with atmospheric composition measurements made using multiple analytical methods. For example, resulting OM and OC concentrations were compared with residual OM and TOR OC concentrations, respectively.

## 2    Methods

Ambient aerosol samples, collected onto Teflon filters from five SEARCH network sites over eight years, were analyzed by FT-IR absorption spectrometry (Sect. 2.1). A series of laboratory standards that mimicked the ambient samples were collected using a range of relevant pure chemicals, and spectra were explored to confirm that molecular environments were atmospherically relevant (Sect. 2.2). FT-IR spectra were acquired (Sect. 2.3), and outliers were detected; these were either set aside during model development or removed from the dataset (Sect. 2.4). Calibration models were developed to measure

five functional groups using multivariate analysis (Sect. 2.5). The resulting calibration models were described by interpreting important spectral variables (Sect. 2.5.2). While no direct measurements for evaluating the functional group model measurements exist, estimates of OM from mass and measured components and TOR OC concentrations were used for comparison, and the van Krevelen space was used to compare other measurements of aerosol composition (Sect. 2.5.2). Method detection limits were applied (Sect. 2.5.3), and uncertainties in model measurements of functional

groups/predictions of OM were estimated (Sect. 2.6).

### 2.1  SEARCH network samples, network data, and field blanks

Aerosol composition in the southeast was characterized from 1999 to 2016 by the SEARCH network. The SEARCH network was unique in that it focused on one region of the US, with sites in urban/rural pairs (Birmingham and Centreville in Alabama; Atlanta and Yorkville in Georgia). Measurement methods were advanced and comprehensive, including real-time

gas phase measurements, light and mass-based measurements of total particles, a variety of particle-phase composition measurements (trace elements, inorganic salts, OC, and elemental carbon), and supporting meteorological variables.

Filter samples of ambient aerosol collected in the SEARCH network from 2009–2016 were used in the present study. The sampling sites included urban Birmingham (BHM) and rural Centreville (CTR) in Alabama, urban Jefferson Street, Atlanta (JST) and rural Yorkville (YRK) in Georgia, and rural Outlying Landing Field (OLF) near Pensacola in Florida (Edgerton et

al., 2005). Samples from collocated samplers at the JST site (cJST) were used to calculate the sampling uncertainty of the




functional group measurements (Sect. 2.6). Three additional SEARCH network sites were closed before 2016 and were therefore not included in the current study; sampling in the SEARCH network ended in 2016, on different dates for each site. Samples analyzed in this work were collected using the Federal Reference Method (U.S. Environmental Protection Agency, 2011). Briefly, Partisol Plus 2025 samplers (Rupprecht & Patashnick/Fisher Scientific, http://www.thermofisher.com/) were

used to collect ambient particulate matter smaller than 2.5 μm aerodynamic diameter ($PM_{2.5}$) at 16.7 liters per minute onto MTL 47 mm PTFE filters with 2 μm pore size (Measurement Technology Laboratories, https://mtlcorp.com/filters). Gravimetric analysis of $PM_{2.5}$ mass and X-ray fluorescence of trace metals concentrations were performed using these filters. Additional filter samples were collected and analyzed by the SEARCH network: 37 mm quartz filters for TOR analysis of OC and elemental carbon concentrations, 47 mm PTFE filters for $SO_4^{2-}$, $NO_3^-$, and $NH_4^+$ analyses, and 47 mm Nylon and

cellulose filters for negative artifact $NO_3^-$ and $NH_4^+$ analyses, respectively (Edgerton et al., 2005). SEARCH TOR OC measurements are blank corrected using annual network-wide mean field blank OC concentrations.

One in three days, seasonally representative (January, April, July, and October) samples from 2009–2015, as well as daily samples from 2016, were analyzed using FT-IR spectrometry. The one in three days sampling schedule matched the sampling for TOR OC measurements. At each site, ~30–45 samples were analyzed by FT-IR spectrometry per year from

2009 to 2015; 1474 ambient sample filters were included altogether in this study. 359 field blank filters were used (approximately two field blank filters per month, per site).

In contrast to other networks, there were some advantages and challenges of SEARCH sampling for FT-IR analyses. Unlike IMPROVE samples, filters were shipped and stored at <4°C (from Aerosol Research and Analysis, Inc., ARA, in Morrisville, North Carolina), to minimize loss of volatile species. Gravimetric filter measurements were made in an

environmentally controlled weigh space to minimize uncertainty in water content (Edgerton et al., 2005), a control technique the IMPROVE network has only recently implemented. However, the loading of SEARCH network filter samples was generally lower than that of the IMPROVE network. While the IMPROVE network uses 25 mm diameter filters and a flow rate of 22.8 LPM, the SEARCH network used relatively large filters (47 mm diameter) and a lower flow rate (16.7 LPM), following the FRM sampling procedures (Mikhailov et al., 2009). The Chemical Speciation Network (CSN) also uses 47 mm

diameter filters for collection, and similarly to SEARCH, filters are shipped and stored cold; however, the SEARCH aerosol loading was higher than that in the CSN, which uses a flow rate of 6.7 LPM and 47 mm diameter filters. In addition, the SEARCH filters were constructed of thicker PTFE material, overlapping some aerosol sample peaks in transmission spectrometry and producing strong, variable FT-IR spectral feature related to scattering by PTFE.

## 2.2 Laboratory standard generation

Laboratory standards used to measure functional group concentrations were produced by collecting particles of pure chemicals onto 47 mm MTL PTFE filters to mimic ambient SEARCH network samples. The aerosol generation system consisted of an atomizer (model 3076 Constant Output Atomizer, TSI Inc.), a custom-built diffusion dryer, and a Partisol





(FRM) aerosol sampler operated at 16.7 LPM. The atomizer was supplied with pure chemical solutions and filtered house air (Model 3074B Filtered Air Supply, TSI Inc., http://www.TSI.com/).

Two types of laboratory blanks were collected. "Chamber blanks" were collected using deionized (DI) water (≥18.2 MΩ purity) for 10–180 minutes or isopropanol (IPA; Spectrum Spectrasolv grade) for 5–35 minutes. "Method blanks" were placed in the aerosol generation system and handled identically to laboratory standards, but the pump was not turned on. One method blank was collected while each pure chemical was being collected. Multiple pure chemicals (Table 1) were chosen to represent each of the organic functional groups calibrated (see Sect. 2.5).



**Table 1. Pure chemicals collected as laboratory standards and used in the calibration of FT-IR spectra for functional group concentrations.**

| Pure Chemical | Chemical Character | Reason for Including in Model | O/C | H/C | OM/OC | Molecular Structure | Molecular Formula |
|---|---|---|---|---|---|---|---|
| Squalene | Unsaturated hydrocarbon | Represents unsaturated hydrocarbons | 0.00 | 1.67 | 1.14 | | $C_{30}H_{50}$ |
| Oxalic Acid | Oxalic acid | Abundant chemical in atmospheric aerosol | 2.00 | 1.00 | 3.75 | | $C_2H_2O_4$ |
| Malonic Acid | Short chain length di-acid | Abundant chemical in atmospheric aerosol | 1.33 | 1.33 | 2.89 | | $C_3H_4O_4$ |
| Succinic Acid | Short chain length di-acid | Mid-range length carboxylic acid | 1.00 | 1.50 | 2.67 | | $C_4H_6O_4$ |
| Suberic Acid | Medium chain length di-acid | High-range length carboxylic acid (spectrum similar to long-chain mono-carboxylic acids) | 0.50 | 1.75 | 1.81 | | $C_8H_{14}O_4$ |
| Terephthalic Acid | Aromatic acid | Represents aromatic acids, especially industrial emissions | 0.50 | 0.75 | 3.67 | | $C_8H_6O_4$ |
| D-Alanine | Amino acid | Amino acid abundant in atmospheric aerosol | 0.67 | 2.33 | 2.47 | | $C_3H_7NO_2$ |
| Ammonium Oxalate | Carboxylate salt | Theoretically atmospherically abundant carboxylate salt | 2.00 | 4.00 | 5.17 | | $C_2H_8N_2O_4$ |
| Sodium Oxalate | Carboxylate salt | Theoretically atmospherically abundant carboxylate salt | 2.00 | 2.00 | 5.58 | | $C_2O_4Na_2$ |





| | | | | | | | |
|---|---|---|---|---|---|---|---|
| D-(+)-Glucono-delta-Lactone | Lactone | Represents cyclic carbonyls, including carbohydrates | 1.00 | 1.67 | 2.47 | | $C_6H_{10}O_6$ |
| Tannic "Acid" | Humic-like substance | Representative of oligomeric substances (carbonyl, phenolic OH) | 0.61 | 0.68 | 1.86 | | $C_{76}H_{52}O_{46}$ |
| Ethyl Palmitate | Aliphatic ester | Representative of esters | 0.11 | 2.00 | 1.32 | | $C_{18}H_{36}O_2$ |
| 10-Nonadecanone | Aliphatic ketone | Representative of ketones | 0.05 | 2.00 | 1.30 | | $C_{19}H_{38}O$ |
| meso-Erythritol | Biogenic tetrol | Abundant product of isoprene oxidation | 1.00 | 2.50 | 2.54 | | $C_4H_{10}O_4$ |
| D-(+)-Glucose | Carbohydrate | Representative of carbohydrates | 1.00 | 2.00 | 2.50 | | $C_6H_{12}O_6$ |
| Levoglucosan | Biomass burning tracer | Tracer of biomass burning emissions | 0.83 | 1.67 | 2.25 | | $C_6H_{10}O_5$ |
| 4-Nitrocatechol | Phenol | Representative of phenols, typical of biomass burning emissions | 0.67 | 0.83 | 2.15 | | $C_6H_5NO_4$ |
| 1-Docosanol | Long chain length alcohol | Representative of fatty alcohols | 0.05 | 2.09 | 1.24 | | $C_{22}H_{46}O$ |



| | | | | | | | | |
|---|---|---|---|---|---|---|---|---|
| Ammonium Sulfate | Interferent | Abundant in atmospheric aerosol (inorganic salt) | -- | -- | -- | | | $(NH_4)_2SO_4$ |
| Ammonium Nitrate | Interferent | Abundant in atmospheric aerosol (inorganic salt) | -- | -- | -- | | | $NH_4NO_3$ |
| Magnesium Chloride, Hexahydrate | Interferent (water) | Does not absorb in infrared region of interest, but strongly hygroscopic so that spectrum represents particle (hydrate and liquid) water | -- | -- | -- | $Cl^-\ Mg^{2+}\ Cl^-$ | | $MgCl_2$ |




For each pure chemical, 10–20 filters of varying masses were collected (for 1–35 minutes); 315 chemical standards was produced. The mass of functional group deposited onto each laboratory standard filter was calculated as the difference in filter mass (in µg) before and after collection. Each filter was pre- and post-weighed at least three times using a high precision balance (±2 µg; model XP2U, Mettler–Toledo, https://www.mt.com). The total quantity of functional group anticipated in ambient

samples, based on literature values, was used to determine the range collected for each chemical. For example, suberic acid standards were generated in the range of 0.04–4 µmol C=O per filter, which is higher than expected for suberic acid itself (Gao et al., 2006), but within the range anticipated for total C=O (Polidori et al., 2008). The range of measured functional group concentrations in ambient samples was also compared to the dynamic range included in the models (Sect. 3.3.2).

Most of the pure chemical solutions were prepared in IPA and/or DI water; a small number were prepared in ethanol (Koptec

Pure Grade). Impurities in the solvents were identified by looking at FT-IR spectra of chamber blanks. However, weights of the impurities in the IPA and ethanol were within the uncertainty of the high precision balance when collected for up to 35 minutes, and were not predictive in the functional group models. No impurities were discovered in the DI water. Sonication for up to two hours was used for some solutions. Concentrations and other details of the pure chemical solutions are listed in the Supplementary Material, Sect. 1–3.

Molecular environments of the laboratory standards were influential to the infrared spectra and were explored qualitatively (observations summarized in Sect. 3.2, and more detail compiled in Supplementary Material). Hydrogen and ionic bonding patterns were interpreted within spectra of collected standards containing single chemicals. In some cases, a chemical was not included in the model due to a variable hydrogen bonding pattern. Multi-component laboratory standards (containing two chemicals per filter) were also generated to qualitatively assess the interactions between molecules of different chemicals. The

influence of humidity on the laboratory standards was assessed by exposing a selection of laboratory standards to a dry and a wet environment (a desiccator with silica beads and a desiccator with water, respectively). Blank filters as well as laboratory standard filters containing a hydrophobic chemical (squalene) were analyzed as controls. Each filter was exposed to each environment for one week.

### 2.3 FT-IR spectrometry analysis: Spectrum acquisition

Analyses of the sample and laboratory standard filters were carried out in transmission mode on a Bruker Tensor II FT-IR spectrometer (Bruker Optics, Inc.; http://www.bruker.com/) equipped with a mid-infrared light source and liquid nitrogen cooled mercury cadmium telluride detector. Each filter was placed into a custom-built (Debus et al., 2018) chamber within the FT-IR spectrometer that was continuously flushed with air scrubbed of $H_2O$ and $CO_2$ (model VCDA air purge system, Puregas, LLC, http://www.puregas.com/; <10% humidity). Additional information about the FT-IR spectrometry analyses can be found

elsewhere (Debus et al., 2018; Ruthenburg et al., 2014). Spectra were collected between 4000 and 420 cm$^{-1}$, but 1500 to 400 cm$^{-1}$ was excluded due to strong PTFE filter absorption (Weakley et al., 2016) and highly variable absorption between chemicals.

Subsets of ambient SEARCH samples were re-analyzed after differing periods to determine whether FT-IR handling/analysis, short-term storage and transport, or long-term storage had substantially affected the spectra. Changes in predicted functional group and OM concentrations over each period were compared to the sampling uncertainty to assess whether a measurable bias

could be observed. The results of this re-analysis demonstrated that: (1) duplicate analyses via FT-IR spectrometry were reproducible (-3% median bias in OM concentrations), and FT-IR analysis did not impact filter sample composition; but (2)



decreases in some functional group concentrations for some samples were measurable within the first year or two after sampling (e.g., approx. -10% yr$^{-1}$ and -5% yr$^{-1}$ median bias in aCOH and OM concentrations, respectively); (3) however samples stabilize in storage and no longer had measurable concentration changes after several years (approx. 5% median bias measured between seven years versus five years after collection). Additional information about the re-analyses is summarized in the Supplementary

Material, Sect. 16.

### 2.4   Outlier detection and handling

Outlier laboratory standards and blanks were identified, and data were removed or set aside during the calibration process so that models were constructed and evaluated based on data with minimal errors. Laboratory standards and blanks with the following characteristics were explored as potential outliers: (1) unusually strong water vapor absorption bands in the spectra; (2)

uncharacteristic and atypical chemical absorption bands in the spectra; (3) atypical molar absorptivities compared to other laboratory standards; or (4) collected material weights that were too high or essentially zero (except for blanks). Spectra with anomalously high leverage values (those which disproportionately impacted the model result; Hoaglin and Welsch, 1978) were also examined.

Ambient samples and field blanks are expected to be occasionally anomalous: for example, filters can be ripped, and field blanks

can be swapped with ambient samples. Potential outlier ambient samples and field blanks were identified using a variety of methods. We treated the confirmed sample outliers in two ways. If no explanation for poor data quality could be determined, the spectrum was set aside into the validation set (see Sect. 2.5) and not used in the model construction process. Functional group concentrations of these spectra were measured and reported after model construction. If an explanation for poor data quality was determined, the spectrum was excluded from the analyses entirely.

We identified and further explored ambient samples with the following characteristics as potential outliers: (1) a spectrum that was visibly anomalous (e.g., swapped with a blank, having a hole, or having strong water vapor absorption bands); (2) a spectrum corresponding to a high TOR OC concentration, but low infrared absorption; or (3) a high error in prediction after calibration. Principal components analysis (PCA), a technique used to find the patterns describing maximum variance in a dataset (Naes et al., 2002), was additionally used to identify potential outliers. Overall, approximately 8% (128/1656) of the

ambient samples were removed from the dataset, and 31 were set aside for later prediction (in the validation set). Samples missing a TOR OC concentration were still included in the results.

### 2.5   Building and evaluating the functional group calibration models

Six functional group calibration models were initially constructed: saturated/aliphatic C-H (aCH), unsaturated C-H (unsCH), carboxylic acids (COOH), oxalate C=O (oxOCO), non-oxalate C=O (noxCO), and alcohol C-OH (aCOH). We used a linear

regression between COOH and noxCO to differentiate between carboxylic C=O and "naCO" (non-acid, non-oxalate/other C=O; see the Supplementary Material Sect. 11 and Takahama et al., 2013). This was necessary because, although the C=O stretching bands of carboxylic acids are theoretically shifted to lower wavenumbers (~1700–1710 cm$^{-1}$) than an unperturbed C=O stretching band (~1725–1740 cm$^{-1}$; Mayo et al., 2003), there is not a clear separation between these two types of C=O and this spectral range is not unique to carboxylic acids. A calibration model for unsCH was developed, but we did not include the values



in our results because a substantial fraction of the samples was below the detection limit (see Sect. 2.5.3). The five functional groups reported are therefore aCH, COOH, oxOCO, naCO, and aCOH.

Although we used literature values as an initial estimate for the range of functional groups in the ambient samples, we further determined the maximum number of moles of each functional group to include in the models using a randomized energy

minimization algorithm called simulating annealing (Ledesma et al., 2012; see Supplementary Material Sect. 8–9 for discussion on this method). The final values determined were: 30 μmol aCH and unsCH, 5 μmol COOH, 4 μmol oxOCO, 4 μmol noxCO, and 10 μmol aCOH.

Partial least squares (PLS) regression was used for calibration and performed in Matlab using the nonlinear iterative partial least squares (NIPALS) algorithm (Wold and Sjostrom, 2001). A mathematical description of PLS regression for functional group

measurement is given in Reggente et al., 2018 and Ruthenburg et al., 2014. Briefly, PLS identifies a set of factors describing the variations in the laboratory standard spectra and known functional group moles based on the maximal covariance between them. The spectral patterns of these factors (loadings) and the respective contributions of the factors to each standard spectrum (scores) are derived. The spectra and moles of functional groups in the calibration set of laboratory standards are mean-centered prior to use in the PLS model. A set of regression coefficients, similar in concept to the slope of a univariate calibration curve, is

calculated from the scores and loadings.

Laboratory and field data were partitioned into subsets for model development and application: (1) a calibration set of standards for training the calibration models; (2) a test set of standards used for testing the model parameters with respect to the response of laboratory standards; (3) a test set of ambient SEARCH samples for testing the model parameters with respect to bulk metrics such as residual OM and TOR OC; and (4) a validation set for evaluating model performance using the final model parameters.

The calibration set of standards contained seven laboratory standards of each chemical, two chamber blanks per chemical, one method blank per chemical (56 total laboratory blanks), and 20% of the available SEARCH network field blanks (52). The test set of standards contained one to 14 standards per chemical (depending on the number of available standards), and the rest of the laboratory blanks (20) and field blanks (307). The test of samples set contained 1125 ambient samples and the same 307 test set field blanks. The validation set of samples contained 318 ambient samples, as well as extreme samples that were identified as

possible outliers, but no explanation for their removal from the dataset was found (31; Sect. 2.4). The test and validation sets of ambient samples were combined for all figures and metrics.

The calibration set for each functional group model contained chemicals as organic "interferents" if the particular molecule did not contain that functional group, with quantities of functional group set to zero. This accounted for spectral overlap between functional groups. For example, carboxylic acids and alcohols were quantified separately, but both functional groups contain an

O-H bond absorbing in a similar mid-infrared range. Changing the number of such organic interferent standards in each model had a negligible impact on prediction.

Each functional group model was tested by applying it to the test set of laboratory standards using Eq. 1. The moles ($n$) of each functional group ($g$) in a laboratory standard ($j$) with spectrum $x_j$ is measured as the sum of inner products as:

$$n_{ig} = \Sigma_{ij} b_{gj} x_{ij} \tag{1}$$

The modeled moles of functional groups in the test set of laboratory standards (Eq. (1)) were plotted against the known moles from filter weights. An orthogonal least squares regression of the moles from the model and filter weights was fitted, and the median error, correlation coefficient, and slope were examined. Model inputs (such as the subset of laboratory standards included





in the calibration versus test sets of standards, and the maximum quantity of chemical in calibration set laboratory standards) were altered to optimize the modeled test set of standards.

Multiple methods were tested to find the optimal number of factors for each SEARCH functional group model (see Supplementary Material Sect. 8). The minimum root mean squared error of cross-validation (RMSECV) with a *k*-fold of 3 was

selected because of its speed and simplicity. Overfitting of the COOH functional group was observed (resulting in overestimation of naCO concentrations). To minimize this effect, the maximum number of factors was constrained to 15 factors for this model. All other functional groups were constrained to 25 factors. The resulting numbers of factors for each functional group calibration model selected by the automated minimum RMSECV method were: 21, 25, 15, 24, 20, and 25 for aCH, unsCH, COOH, oxOCO, noxCO, and aCOH, respectively.

### 2.5.1 Bulk OC and OM concentration estimates

The concentration of OC in each ambient sample, $OC_i$, was estimated as the sum of measured C atoms ("functional group OC"), assuming the following C atom contributions per functional group ($\lambda_g$): aCH=0.5C, COOH=1C, oxOCO=1C, naCO=1C, and aCOH=0.5C (Eq. (2)). The same values were used by Russell et al. (2003). For the four functional groups measured by Ruthenburg et al, 2014, the same assumptions were made except that aCOH was assumed to contribute no C atoms. In Eq. 2, the

moles of functional group $g$ in the $i$th sample are denoted $n_{ig}$, and 12.011 g mol$^{-1}$ is the molar mass of C:

$$OC_i = 12.01 \Sigma_g n_{ig} \lambda_g \qquad (2)$$

These assumed values of $\lambda_g$ therefore influence the predicted functional group OC concentrations. The values of $\lambda_g$ are supported by parallel measurements and modeling (Takahama and Ruggeri, 2017) as well as Monte Carlo simulations (forthcoming work by Burki et al.). Similarly, OM concentrations were calculated from summed functional groups including the same assumptions

for C contributions, plus all associated O and H atoms.

The OM/OC ratio was calculated by dividing the summed OM concentrations by the summed OC concentrations. Although TOR OC concentrations have been suggested for normalizing OM/OC ratios in the past (Reggente et al., 2018), the summed OC concentrations were used because these two values give a consistent representation of organic composition than a ratio between an FT-IR spectrometry measurement and TOR measurement (these techniques capture slightly different portions of organic

species/functional groups).

### 2.5.2 Model evaluation: Interpretation of model predictors and comparison with external measurements

The variable importance in the projection (VIP) scores were calculated to simplify interpretation of the variance described by the calibration models. VIP scores have been previously utilized to demonstrate the importance of predictor variables (here, absorbance at each wavenumber) in PLS when the predictor variables are not independent (Chong and Jun, 2005). This applies

to the current method because in infrared absorption spectra, absorbance at separate wavenumbers varies together (bonds can absorb in multiple regions simultaneously). Essentially, the VIP scores describe the relative importance of each wavenumber in the model by taking into account the *y*-variance (functional group quantity) explained by the model as weighted onto each PLS factor. The models were evaluated using the VIP scores by determining whether the important (and unimportant) wavenumbers



in the models corresponded to known functional group absorption bands expected in ambient aerosols. See the Supplementary Material Sect. 12 for the equation used to derive the VIP scores for the total functional group OM.

Reference measurements to validate functional group concentrations directly do not exist: our measurements represent the first time these functional groups have been quantified in southeastern US aerosol samples to our knowledge. Instead, we evaluated

our predictions against residual OM and TOR OC. The residual OM was calculated by subtracting the weighted sum of the major inorganic chemical constituents and elemental carbon from $PM_{2.5}$ mass for each SEARCH sample using the equation described by Hand et al. (2012b). A particle water correction was made (Dabek-Zlotorzynska et al., 2011; Simon et al., 2011). Metrics used between measured and reference OM or OC were coefficient of determination $R^2$, bias-corrected error (also known as the median absolute deviation, as previously described by Weakley et al., 2016), and orthogonal least squares regression slope. Because the

mass recovery was expected to be less than 100%, bias was not a relevant metric. The 95% confidence intervals were calculated around the regression slope by bootstrapping. The regression slopes and confidence intervals gave an estimate of the mass recoveries of OM and OC, relative to each reference method.

Another method for evaluating the model performance was comparing the data in a van Krevelen diagram to aerosol mass spectrometry data collected in the southeastern US. A van Krevelen diagram describes the overall elemental composition of OM

in the two dimensional space of atomic H/C versus atomic O/C. It should be noted that the material collected in the SEARCH network is $PM_{2.5}$, and not $PM_1$, as measured using aerosol mass spectrometry; however, the difference in sources contributing to OM between the two fractions may be small (Schum et al., 2018). In the future, $PM_1$ measurements could be considered for studying the comparison of FT-IR spectrometry and aerosol mass spectrometry measurements.

### 2.5.3    Method detection limits

The method detection limit (MDL) of each functional group concentration was estimated as three times the standard deviation of all laboratory and field blank functional group concentrations measured in the test set of standards. The MDL of the functional group OM and OC concentrations were estimated as the root of the sum of squares of the blank OM and OC concentrations predicted in the test sets. No samples were excluded from the results or plots based on the OM or OC MDLs. All ambient sample functional group concentrations predicted below the corresponding MDL were replaced with the value of MDL/2. This censoring

technique has been applied in the past for multivariate analysis of environmental data (Polissar et al., 1998). When data were left un-censored, some values were negative, and therefore ratios such as the OM/OC were misrepresented. Thus, although censoring of environmental data has obvious drawbacks (Helsel, 2005), the MDL/2 replacement and use of robust metrics such as median and percentiles were determined to provide the most accurate summary data. Samples with three or more functional group concentrations below the respective MDLs were not included in the O/ C and H/ C ratios (used in the van Krevelen diagram).

This was done because in these cases, the ratio was dominated by only one or two functional group contributions, and appeared as a straight line of datapoints on the van Krevelen diagram (and was not informative). These samples were, however, left in the dataset for all other figures and metrics so that the data were not biased toward higher functional group/OM concentrations.





### 2.6 Model uncertainties

The precision of the functional group measurement method was evaluated using two approaches, which attempted to evaluate some of the most substantial potential sources of uncertainty in the method. The first approach was the comparison of functional group concentrations measured from two collocated sampling sites within the SEARCH network ("sampling uncertainty"). The

second approach was the calculation of confidence intervals (bootstrapped) around the functional group concentrations measured using a set of 18 model predictions, each of which had one organic chemical standard removed from the models ("chemical selection uncertainty").

The sampling uncertainty accounted for the sensitivity of the FT-IR spectrometry analysis procedure to differences in filter substrates, FT-IR analysis handling, and SEARCH network sampling and handling procedures. Sampling uncertainty was

calculated (Hyslop and White, 2008, 2009) using measured functional group concentrations from the JST site and its collocated site, cJST. The collocated sampler was used to collect $PM_{2.5}$ for only a subset of dates (2009–2011, October 2015, and 2016). This uncertainty was used throughout the study as the most complete estimate of method uncertainty, since it included most possible sources of uncertainty, aside from those arising from selection model inputs and parameters.

The chemical selection uncertainty accounted for the possible impact of excluding a particular atmospherically important

chemical from our models, *within the bounds of our chemical list.* We performed this "leave one out" analysis with the expectation that the sensitivity of the models would be similar between chemicals in the current models as well as some hypothetical, atmospherically important chemicals not included in the models. The chemical selection uncertainty was calculated as follows. Eighteen sets of models were constructed, each excluding one organic chemical. Ambient functional group concentrations were measured using all models. For each functional group, the concentrations measured by all models were

aggregated into one vector. The uncertainty over all samples and models was then determined using the sampling uncertainty equations between the "base case" concentrations (calculated with all chemicals included) and the "leave one out" concentrations.



## 3 Results and discussion

In the results that follow, we highlight how we addressed the multiple challenges of developing robust calibration models for measuring functional groups and OM concentrations in SEARCH ambient samples. In Sect. 3.1, the selected set of atmospherically relevant laboratory standards and the functional groups quantified are discussed. In Sect. 3.2, issues of molecular

environment are qualitatively evaluated, including assessing humidity impacts and particle water absorbance. The accuracy of the models is dependent on model parameters and inputs; the model results were evaluated in Sect. 3.3 by confirming that predictive model spectral features are atmospherically relevant and predicted laboratory standards concentrations were accurate. Although functional group and OM concentrations cannot be directly compared to external (other method) measurements, Sect. 3.4 highlights comparisons used to evaluate, and provide additional confidence in, the model outputs. These include the fraction

of OM quantifiable considering the portion that absorbs in the modeled spectral region (mass recovery), OM/OC ratios, and a van Krevelen diagram. Sect. 3.5 summarizes some additional uncertainties in the model and future work needed to address these.

### 3.1 Chemicals used in the calibration models to concisely represent atmospheric composition

Known atmospheric OM molecules are comprised mainly of a small number of functional groups, which include C-H, alcohol O-H, and various forms of C=O groups. Relevant C=O groups include carboxylic acids and carboxylates as well as esters,

ketones and lactones. Multiple molecules that contain each of five important functional groups (aCH, COOH, oxOCO, naCO, and aCOH) were included in the calibration models in this work. Selections were made based on the known presence of a molecule in atmospheric OM, or because the molecule exemplified the spectra of a functional group (Figure 1; Table 1). Mass contributions from organic S and N compromise a smaller portion of ambient OM (Liu et al., 2009; Stone et al., 2012) and were not included in this work. Substantial contributions of organosulfates to southeastern aerosol composition are possible

(Hettiyadura et al., 2014), and the OM/OC ratios of small organosulfate molecules are high, therefore, future models may consider such chemicals. The following paragraphs outline the atmospheric relevance and spectral features of each functional group reported in the current models.



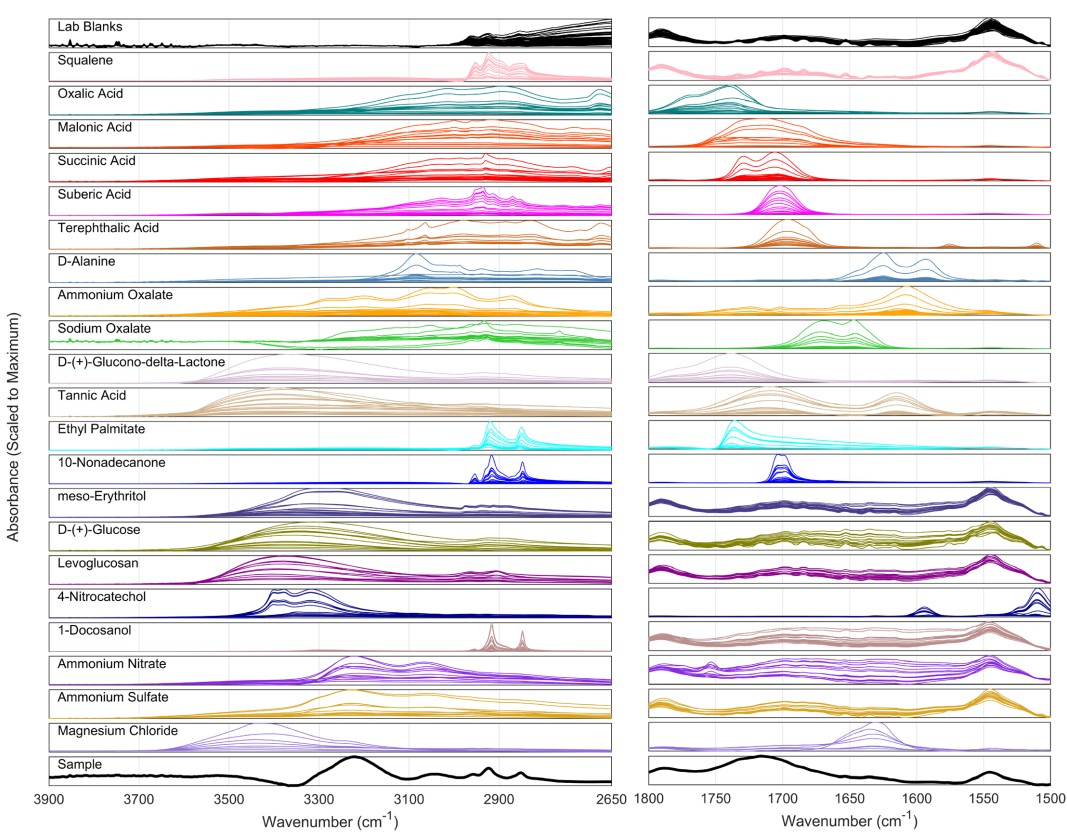

**Figure 1.** FT-IR spectra of all laboratory standards (calibration and test sets). The C-H, O-H, and N-H stretching region was plotted separately (left) from the C=O and C=C stretching region (right). An example ambient SEARCH sample spectrum is plotted for comparison (bottom subplot; 13th Oct 2013 from Birmingham, AL). Spectra are baseline corrected via smoothing splines (Kuzmiakova et al., 2016) and each subplot is scaled to the maximum absorbance for each wavenumber range.



The C-H bond is ubiquitous in atmospheric organic molecules and is present in nearly all chemicals in the models (Table 1; Figure 1). Fresh atmospheric emissions often contain abundant C-H bonds (e.g., alkanes from industrial and biogenic sources; Rogge et al., 1993), although C-H as a functional group should not be attributed only to fresh emissions since it is also plentiful in oxidized material (Schum et al., 2018). Although minor in comparison to C=O or O-H stretching bands, there are some

variations in the C-H stretching bands (e.g., -CH$_2$- at 2926 asymmetric and 2853 symmetric ±10 cm$^{-1}$ for straight-chain alkanes, or 3085–2927 cm$^{-1}$ asymmetric and 3028–2854 cm$^{-1}$ symmetric in cyclic molecules; Mayo et al., 2003). A variety C-H bonds were therefore selected for the models, including ring structures, short-chain and long-chain molecules (additional insight on the variation in C-H bond absorption will be discussed in forthcoming work by Yazdani et al.).

Saturated and unsaturated C-H bonds were quantified separately to distinguish between any differences in sources (e.g., Moretti

et al., 2008). However, the concentrations measured using the unsCH model were not reported because a majority of sample concentrations measured were below the unsCH MDL. These measurements are realistic: low unsCH compared to aCH concentrations have also been observed in work using nuclear magnetic resonance (Moretti et al., 2008) as well as in previous FT-IR functional group calibration work (Guo et al., 2015; Liu et al., 2012; Russell et al., 2009b, 2011). Observed absorption coefficients of unsCH bonds were also low, consistent with theory (Mayo et al., 2003).

Carbonyls are a particularly informative functional group in infrared spectra of ambient OM due to their strong absorption coefficients and high abundance in the atmosphere. A strong, broad C=O stretching band at ~1700–1800 cm$^{-1}$ is observed in ambient OM spectra (Takahama et al., 2013). In particular, molecules containing carboxylic acids may contribute the majority of OM mass (Decesari et al., 2007), the most abundant of which are typically the C$_2$–C$_4$ dicarboxylic acids (Kawamura and Bikkina, 2016). Six carboxylic acids were included in the calibration models. As in our previous work (Ruthenburg et al., 2014),

malonic (C$_3$ dicarboxylic acid) and suberic acids (C$_8$ dicarboxylic acid) were included. The latter represents longer-chain carboxylic acids because it is spectrally similar to C$_{16}$ and C$_{18}$ monocarboxylic acids (National Institute of Advanced Industrial Science and Technology). Oxalic (C$_2$ dicarboxylic) and succinic (C$_4$ dicarboxylic) acids, which are often the most abundant organic species quantified in OM (Kawamura and Bikkina, 2016), were added to the current models. As a representative aromatic carboxylic acid, terephthalic acid was selected, originating from oxidation of burning plastics or other industrial

activities (Wang et al., 2012). D-alanine, an amino acid, was also included.

Amines and amino acids have been studied in functional group calibrations (Kamruzzaman et al., 2018; Liu et al., 2009). Amines and amino acids could contribute ~5–10% of organic aerosol concentrations, and come from a variety of anthropogenic and biogenic sources (Russell et al., 2011). D-Alanine was included in the current models, differing from other carbonyl-containing spectra by its down-shifted carboxylic C=O stretching band (due to the electron donating power of the adjacent N atom) and C-N

stretching band in the same spectral region. These amino acid bands overlap with those of carboxylate C=O, but multivariate regression factors can account for other features of these functional groups to distinguish between them.

Although southeastern ambient aerosol particles may be acidic (Guo et al., 2015), concentrations of oxalate exceeding those of oxalic acid have been observed in ambient samples (Yang and Yu, 2008). Ammonium and sodium oxalates were therefore included in the models as example carboxylate salts, calibrated separately from the carboxylic acid functional group as oxalate

carbonyl (oxOCO). The FT-IR spectra of ammonium and sodium oxalates were different from the spectra of oxalic and most other carboxylic acids in that the C=O stretching bands of carboxylate salts are below 1700 cm$^{-1}$ (Figure 1). Longer chain carboxylates such as succinates could contribute additional OM. The spectrum of ammonium oxalate also contains two N-H





stretching bands overlapping with O-H stretching bands and carboxylic sum tones at 3500–3100 cm⁻¹, allowing the models to account for ammonium carboxylate interferences to measured COOH and aCOH concentrations.

Other, non-acid (and non-oxalate) carbonyls including esters, cyclic esters, and ketones could be abundant in atmospheric aerosol. Cyclic esters (lactones) within large, multifunctional molecules have been observed in ambient aerosol (Kahnt et al.,

2018), and oxo-carboxylic acids such as cis-pinonic and pyruvic acids are frequently observed in ambient OM (Kawamura and Bikkina, 2016). These non-acid carbonyls were quantified in the models as "naCO", separate from carboxylic acids and oxalate, as in the work of Russell and co-workers (Frossard and Russell, 2012). The naCO was expanded in the present models to include not only a long-chain ketone and ester, but also a lactone (D-glucono-delta-lactone) and a large, conjugated ketone-containing molecule (tannic acid). An aldehyde-containing molecule was also tested in the models, but the solubility of the particular

chemical used (divanillin) limited the maximum mass collected onto filters (also see Supplementary Material, Sect. 4). The lactone was chosen to represent cyclic carbonyl structures such as carbohydrates and furanones (Hamilton et al., 2004). Tannic acid was included in the naCO functional group to represent larger, humic-like molecules. Its spectrum is characterized by a broad C=O stretching band due to the movement of electrons through its multi-ring, oxygenated aromatic structure (Figure 1; Table 1), similar to spectra of observed atmospheric humic-like material (Chen et al., 2016). The molecule is large relative to

atmospheric components observed using typical ion and gas chromatography methods (Gao et al., 2006), and has a known chemical structure (unlike other humic-like candidate molecules). Note that tannic acid contains no COOH moieties, but instead contains ester and ketone naCO, unsCH, aCH, and phenolic aCOH.

Along with non-acid carbonyls, alcohol OH (aCOH) is often recognized as an intermediate within oxidation schemes because the C atom is not maximally oxidized (Heald et al., 2010). A variety of alcohol-containing molecules were included in the models,

typified by broad hydrogen bonded O-H stretching bands ~3500–3100 cm⁻¹ (Figure 1). meso-Erythritol was included as a representative isoprene oxidation product, which is understood to be important in the southeast (Claeys et al., 2004). Phenols were represented by 4-nitrocatechol, which is most often associated with biomass burning and pesticide emissions (Harrison et al., 2005; tannic acid also contains phenol). Along with 4-nitrophenol, levoglucosan is an abundant tracer for biomass burning emissions (Mayol-bracero et al., 2002); it was also included in our previous work (Ruthenburg et al., 2014). Glucose was

spectrally similar to levoglucosan, but was included to represent carbohydrates from other sources such as fungal spores (Caseiro et al., 2007). Although there is little literature discussing long-chain alcohols in atmospheric aerosol, they are indeed present at low quantities (Rogge and Hildemann, 1994). 1-docosanol was therefore included, as in the work of Ruthenburg et al. (2014).

Three types of interferent molecules were included in each of the functional group models: inorganic salts, particle water, and interfering organic species. Ammonium nitrate and ammonium sulfate are abundant in atmospheric aerosol, and overlap

spectrally (N-H stretching) with strongly absorbing organic molecule features, such as O-H and C-H stretching bands. Therefore, these inorganic salts were included as interferents in the models (ammonium nitrate was not included in the Ruthenburg et al., 2014 models). Water also contributes some O-H stretching to aerosol spectra (Frossard and Russell, 2012), so a hygroscopic inorganic salt with negligible inorganic absorption stretches, magnesium chloride (MgCl₂), was also included in the models. To our knowledge, particle water has not been previously accounted for as an interferent in functional group measurements by PLS

calibration of FT-IR spectra, although water interference has been discussed and explored in peak fitting calibrations (Faber et al., 2017; Frossard and Russell, 2012). The result of including particle water as an interferent in the calibration models (collected as MgCl₂) was to contribute spectral features to the models associated with particle water (demonstrated in the VIP scores; see Supplementary Material, Sect. 12). No substantial changes in the measured functional group concentrations (including that of



aCOH), or the predicted OM concentrations, were observed due to the inclusion of particle water standards (Supplementary Material, Sect. 6). This is probably because the humidity in the FT-IR sample chamber is low (0–10%). Particle water was therefore limited to liquid water in un-effloresced highly hygroscopic particles, hydrate water, or as embedded water under aerosol material layers (Frossard and Russell, 2012).

### 5  3.1.1   Addressing uncertainty in model chemical selection

While the chemicals used in the calibration models were selected carefully, using current literature of atmospheric composition, such a concise list will inherently bring about some uncertainty. To demonstrate the robustness of our models to chemical selection, we examined the effect of leaving one chemical at a time out of our calibration models (see Sect. 2.6 and Supplementary Material, Sect. 18). The resulting precision related to chemical selection was within the same range as that

calculated for sampling uncertainty (10–30% bias in median functional group concentrations). The greatest change in predicted functional group concentrations was observed for oxOCO: when either ammonium or sodium oxalate was left out during model construction, the oxOCO model was not robust to the change. This was likely due to the small number of chemicals included in the functional group model (only two) and enhanced by the difference between the spectra of these two chemicals, which contained broad features that overlapped with those of other functional groups. The predicted median OM concentration

decreased by ~25% when oxOCO was not included as a functional group in the models, a change that was attributed not only to the influence of these two, spectrally distinct standard chemicals, but also to the influence of oxOCO standards as "interferents" in models of other functional groups. Interpretation of the predictive spectral features (VIP scores; see Sect. 3.3.1) suggested that the spectral features of oxOCO that overlap with those of other functional groups, when unaccounted for in the models, obscured those features from being fully captured by the models. Thus, by including the additional spectral information of

oxOCO standards as interferents in the other functional group models, other functional groups were more fully and clearly measured.

### 3.2  Molecular environment considerations

Aspects of the environment within and around collected standard particles were examined to discern whether the conditions were relevant to simulated ambient aerosol samples. In particular, three types of molecular-level interactions with the particles of the

collected laboratory standards were considered: (1) hydrogen bonding patterns within pure chemicals; (2) hydrogen and ionic bonding within mixtures of two different chemicals; and (3) changes of pure chemicals due to exposure to water.

The organization and orientation of polar, organic molecules within solid particles is dictated in part by the inter- or intra-molecular hydrogen bonding interactions between H and O atoms (and possibly other electronegative atoms). These hydrogen bonding patterns can strongly influence infrared spectra, causing splitting, broadening, or frequency shifts in absorption bands

(Davey et al., 2006). Dimeric or polymeric hydrogen bonding structures of carboxylic acid standards in the present work were confirmed by the broad O-H stretching band between approximately 3200 and 2600 cm[-1] (with overlaid sum tone absorption bands) and the presence of out-of-plane O-H wagging bands between 950 and 850 cm[-1] (Mayo et al., 2003). Similar hydrogen bonding O-H stretching bands were observed for most alcohols, at higher frequencies due to their weaker hydrogen bonding than carboxylic acids (Mayo et al., 2003). If O-H bonds are unassociated with other polar groups, free O-H stretching peaks are




present in an FT-IR spectrum (Davey et al., 2006; Mikhailov et al., 2009). This was observed in the standards of single, pure chemicals containing multiple polar, oxygenated functional groups, including tartaric acid (not included in models; see discussion and spectra in Supplementary Material, Sect. 4) and 4-nitrocatechol spectra. Free O-H stretching bands were not clearly observed in the SEARCH ambient sample spectra but could have contributed low absorbance within the sample mixture.

Hydrogen bonding within the likely amorphous solid structures of ambient particles (Mikhailov et al., 2009) and the dimeric or polymeric polar, protic chemical used in the calibration was generally consistent.

Laboratory standard filters used for (quantitative) calibration included one chemical on each filter, but hydrogen or ionic bonding interactions between the many chemicals in ambient aerosol samples were expected. We therefore generated laboratory standards with mixtures of pure chemicals, including combinations of two carboxylic acids (malonic with terephthalic acid, malonic acid

with succinic acid), carboxylic acids with alcohols (succinic or malonic acids with *meso*-erythritol, and malonic acid with levoglucosan), and an inorganic salt with carboxylic acids (ammonium nitrate with terephthalic acid, succinic acid, or malonic acid). Hydrogen bonding interactions were observed (see Supplementary Material, Sect. 7, Fig. S-8) in some mixtures, such as those of *meso*-erythritol with malonic and succinic acids, which resulted in the splitting or broadening of the O-H stretching band of *meso*-erythritol (likely due to the formation of additional hydrogen bonding environments). No substantial changes to the C=O

stretching bands were observed. No interactions were visible in the FT-IR spectra between inorganic salts and carboxylic acids, or between some of the mixed polar, protic species, such as malonic acid with terephthalic acid. Oxalate standards, as discussed earlier in Sect. 3.1, accounted for carboxylate salt (ionic bonded OM) contributions to OM concentrations. Based on these observations, models constructed with pure chemical standards could be misattributing some spectral features (adding some error/scatter or bias via over- or under-prediction), but seemingly for only some functional groups and some molecular

interactions.

The influence of water exposure on laboratory standards was examined to demonstrate possible differences between ambient and laboratory generated particles. Although chemical effects were anticipated, including addition of water to non-acid carbonyl groups to form gem-diols or changes in hydrogen bonding structure after deliquescence, there was no spectral evidence of either. Instead, an irreversible decrease in laboratory standard infrared absorption occurred when hygroscopic species were exposed to

humid conditions (glucose, ammonium sulfate, and pyruvic acid; see Supplementary Material, Sect. 6 and Fig. S-5). This was likely the result of a redistribution of collected material away from the infrared beam: there was no consistent and significant change in the weight of standard filters, and a similar decrease in infrared absorption was not observed for the hydrophobic species squalene. Some additional water vapor absorption was also observed. The dry (~0–10% relative humidity) environment of the FT-IR spectrometer sample chamber was also examined by exposing laboratory standards to a dry environment; no effect

on the spectra was observed. Particle water laboratory standards (MgCl$_2$) were included in the calibration models, as described earlier in Sect. 3.1, and effectively accounted for the known portion of by particle water (Dabek-Zlotorzynska et al., 2011; Faber et al., 2017).



### 3.3 Evaluation of model performance

#### 3.3.1 Predictive features of laboratory standards found in the models

We first evaluated model performance by interpreting the spectral features in the models used to measure each functional group. Variable Importance in the Projection (VIP) scores of the predicted OM, shown in Figure 2, demonstrate the predictive spectral

features from the laboratory standards (see Supplementary Material, Sect. 12, for calculation, including method for weighted summing of functional group contributions to VIP scores). A value of one was chosen as a threshold for significant VIP scores, after Chong and Jun (2005) and Weakley et al. (2016).

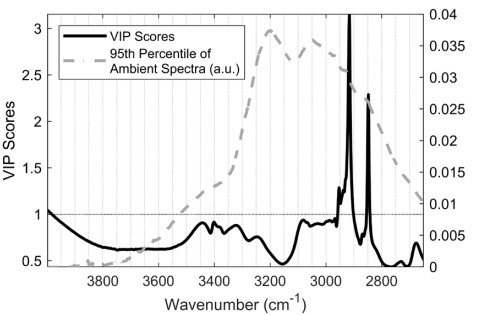 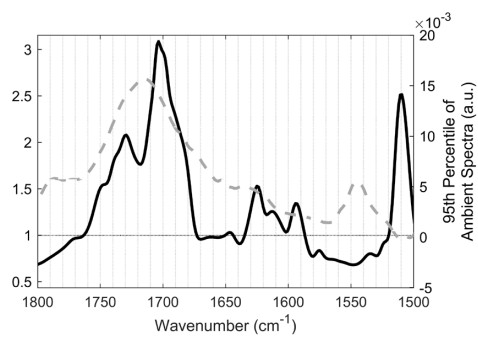

**Figure 2. Variable Importance in the Projection (VIP) scores generated from the calibration models, as a weighted sum of the**
**functional groups (see Supplementary Material, Sect. 12, for calculation). An ambient spectrum is overlaid for comparison (95th percentile of all ambient spectra).**

Spectral features in the VIP scores matched those in absorbing ambient OM but were absent where inorganic species or Teflon absorb in the ambient sample spectra, despite the thick filter material and low aerial density of SEARCH aerosol samples relative

to IMPROVE samples (used in Ruthenburg et al., 2014). This suggests that the list of chemicals assembled and used in the present calibration models did approximate atmospheric composition (a challenge outlined in Sect. 1.3). In addition, including inorganic interferents such as ammonium nitrate and ammonium sulfate successfully allowed the models to avoid accounting for the related spectral features as OM. Predictive features, as determined using the VIP scores and described below, include absorption bands associated with non-acid carbonyls, carboxylic acids, oxalates, D-alanine, alcohols, methylene C-H, and

unsaturated C=C bonds.

Among the most prominent of the features in the VIP scores are several oxygenated functional group bands. Significant VIP scores (>1) are observed in the carbonyl stretching region, corresponding to overlapping absorption bands in the laboratory standards of ethyl palmitate, D-(+)-glucono-delta-lactone and oxalic acid at ~1700 cm[-1], while at 1735 cm[-1], malonic and succinic acids, D-(+)-glucono-delta-lactone, tannic acid and ethyl palmitate may contribute variance. Features specifically

associated with D-alanine (~1625 and 1590 cm[-1]) and with oxalates (sodium oxalate at 1650 cm[-1] and ammonium oxalate at 1610 cm[-1]) were observed at the low end of the C=O stretching region. Two bands with ambiguous interpretation were observed at ~3090 cm[-1] and ~3000 cm[-1]. These two peaks could be associated with malonic acid O-H stretching and overlaid sum tones or with the N-H bonds of ammonium oxalate and/or D-alanine. Between 3410 and 3310 cm[-1], the distinct, sharp O-H stretching features of the 4-nitrocatechol spectra were likely predictive and were observed near a VIP score of one.



Aliphatic and unsaturated carbon backbone features were identified among the significant spectral characteristics, based on laboratory standard spectral features. The asymmetric methyl (2950 cm⁻¹), as well as asymmetric and symmetric methylene (-CH₂-; 2920 cm⁻¹ and 2850 cm⁻¹) stretching bands were prominent, as was the C=C aromatic bending band (1510 cm⁻¹).

5 Several features in the ambient spectra (demonstrated here as the 95[th] percentile of ambient SEARCH spectra, grey dashed trace in Figure 2) were not visible in the VIP scores, indicating that they were not predictive for OM. For example, the symmetric N-H stretching peaks of inorganic (and possibly carboxylate) ammonium at ~3200 and ~3050 cm⁻¹ are visible in the ambient spectral trace, but not the VIP scores. Likewise, the fine water vapor absorption features above 3400 cm⁻¹ and the PTFE absorption features at ~1780 cm⁻¹ and 1545 cm⁻¹ in the ambient spectra were not predictive. Note, however, that the sloping baseline above ~3900 cm⁻¹ was predictive, which could indicate that light scattering by the particulate material on each filter (Weis and Ewing,
10 1996) was a predictive feature of the laboratory standards.

### 3.3.2 Summary of functional group calibration model metrics

Selecting appropriate model parameters based on current understanding was a major challenge in the current study, and was addressed through various model iterations and considerations. The final model parameters and metrics of the results for the five functional groups reported (aCH, COOH, oxOCO, naCO, and aCOH) are summarized in Table 2.

15 .

**Table 2. Summary of calibration model parameters and outputs. Functional groups calibrated, but not reported in the final models, are also included, below the first horizontal line.**

| Functional Group | Method | Dynamic Range (µg m⁻³)¹ | Ratio (λ)² | Num. Chems.³ | Factors (RMSECV) | Standards Test Set Coef. of Det. (R²) | MDL (µg m⁻³) | Percentage of Ambient Samples Above MDL (%) | Median Concentration in Samples (µg m⁻³) | Sampling Uncertainty (µg m⁻³, %)⁴ |
|---|---|---|---|---|---|---|---|---|---|---|
| Saturated Hydrocarbon (aCH) | Calibrated | 0.002 to 1.2 | 1 | 13 | 20 | 0.99 | 0.26 | 94 | 0.90 | 0.15, 16% |
| Carboxylic Acids (COOH) | Calibrated | 0.04 to 3.3 | 1 | 6 | 15 | 0.98 | 0.26 | 84 | 0.63 | 0.15, 28% |
| Oxalate Carbonyl (oxOCO) | Calibrated | 0.07 to 0.65 | 1 | 2 | 23 | 0.93 | 0.04 | 99 | 0.27 | 0.04, 18% |
| Non-Acid Carbonyl (naCO) | Partitioned | -- | 1 | -- | -- | -- | 0.04 | 92 | 0.25 | 0.08, 26% |
| Alcohol (aCOH) | Calibrated | 0.04 to 7.0 | 0.5 | 7 | 25 | 0.98 | 0.24 | 88 | 0.60 | 0.13, 25% |
| Unsaturated Hydrocarbon (unsCH)⁵ | Calibrated | 0.002 to 0.39 | 0.5 | 4 | 25 | 0.99 | 0.08 | 12 | 0.04 | 0.03, 21% |
| Non-Oxalate Carbonyl (noxCO) | Calibrated | 0.04 to 2.6 | -- | 10 | 19 | 0.98 | 0.10 | 99 | 0.64 | 0.04, 18% |
| Organic Matter (OM) | Predicted as sum | -- | -- | 20 | -- | -- | 0.45 | 80 | 2.1 | 0.38, 14% |
| Organic Carbon (OC) | Predicted as sum | -- | -- | 20 | -- | -- | 0.25 | 81 | 1.0 | 0.19, 14% |

**¹ Dynamic range of the standards included for each functional group, as well as the Num. Chems., factors, and standards test set coef. of det. (R²), could only be tabulated for calibrated functional groups. The concentrations are estimated based on the volume of air**
20 **collected at 16.7 LPM for 24 hours.**
**²The ratio used in summing to OC is the ratio of the number of C atoms per functional group, represented as λ.**





[3] The number of chemicals ("Num. Chems.") corresponds to the number of pure chemicals that contained the particular functional group.
[4] Values are reported with significant digits determined based on the sampling uncertainty (last column) and the number of significant digits afforded by the high precision balance used to weigh the lab standard filters.

[5] Most unsCH concentrations in ambient samples were below MDL, and were not reported or used in predicting OM, OC concentrations

As noted in Sect. 2.5, the naCO concentrations could not be calibrated because of spectral overlap and were instead determined by partitioning excess noxCO relative to COOH concentrations (Supplementary Material Sect. 11). The final two rows of the

table give the prediction metrics for OM and OC, which were derived from the five reported functional groups (see Sect. 2.5). The dynamic ranges of laboratory standards used in each functional group model were inclusive of, and similar to, the range of concentrations measured within the ambient samples: for example, aCH concentrations ranged from 0.002 to 1.2 μg m$^{-3}$ in laboratory standards, and from 0.02 to 0.46 μg m$^{-3}$ in the samples (1$^{st}$ to 99$^{th}$ percentiles of sample concentrations). This demonstrated the success of using previous literature and simulated annealing to select the maximum functional group

concentrations in the models (see Sect. 2.5), one aspect of the major challenges anticipated in this work. Likewise, other parameters such as the number and type of non-interfering chemicals included in each model (listed in the "Num. Chems." column of Table 2), as well as the number of PLS model factors, were explored in depth by examining the atmospheric likelihood of results when iterating manually over those parameters. As described in Sect. 2.5, many methods for selecting the number of PLS factors were tested, and RMSECV was used because of its flexibility and simplicity.

The correlations between the functional group moles measured via FT-IR spectrometry and gravimetric analysis for laboratory standards were strong: R$^2$≥0.93 for all calibrated functional groups. Normalized errors in prediction for the test sets were 7–16%, and slopes of the cross-plots were 0.91–1.05 for all calibrated functional groups (see Supplementary Material, Figs. S-13 and S-14). As expected, some ambient sample functional group concentrations were below MDLs. However, for all functional groups, the median concentration measured in the ambient samples was greater than the MDL (Table 2). The predicted median

concentrations of OM and OC in the ambient samples were well above the respective MDLs and ~80% of the ambient sample predicted concentrations were greater than the MDLs in both case. Note that the values discussed in this paragraph were calculated before the censoring of the data below functional group MDLs, as discussed in Sect. 2.5.2.

Sampling uncertainty (Sect. 2.6) was 14% (0.39 μg m$^{-3}$) for OM and 14% (0.19 μg m$^{-3}$) for OC (Table 2). These low sampling uncertainty values demonstrated that: (a) the filter sampling, handling and storage methods were reproducible; and that (b) the

functional group calibration procedures were reproducible.

### 3.4 Using aerosol composition to evaluate FT-IR functional group measurements

There are scarce measurements of functional groups to validate the FT-IR/PLS method developed here. We address this challenge by instead corroborating our measurements with multiple qualitative and quantitative metrics from separate methods. We evaluate the model results by comparing to bulk measurements including residual OM concentrations, TOR OC

concentrations, and ratios of OM/OC, O/C and H/C from other techniques. Expected trends between urban/rural pairs, seasons, and functional groups, based on previous research, were also used.





### 3.4.1    Evaluating FT-IR measurements: Mass recovery

The concentrations of functional group OM and OC were not expected to be 100% of the actual concentrations in ambient samples because some bonds do not absorb mid-infrared light within the modeled range (4000–1500 cm$^{-1}$). For example, squalene ($C_{30}H_{50}$) contains five C atoms per molecule with only C-C bonds; those five C atoms do not absorb in the mid-infrared

range of interest, so squalene OC concentrations will be underestimated by 17%. Similarly, levoglucosan ($C_6H_{10}O_5$) contains two O atoms within its rings; since the stretching region of C-O is at 1300–1000 cm$^{-1}$ (Pavia et al., 2009), where PTFE also absorbs, functional group OM will underestimate levoglucosan OM by ~20%. Thus, in the prediction of OM or OC in ambient samples, a "mass recovery" of approximately 70–80% OM or OC was expected (Takahama and Ruggeri, 2017). In addition, since the composition of ambient samples vary, the mass recoveries were expected to differ, and thus to add scatter to the estimated

functional group OM and OC concentrations. Quantifying a substantial fraction of the OM (and OC), despite the lack of mid-infrared absorption of some relevant molecular bonds, was a major challenge in this current work, addressed by chemical and model input selection.

Other methods of OM or OC characterization are understood to have a mass recovery below 100%. Similar to FT-IR/PLS, the mass recovery of organics in aerosol mass spectrometry is 75% for O/C ratios and 91% for H/C ratios (Aiken et al., 2008;

assuming constant collection and relative ionization efficiencies with particle composition). Although TOR OC mass recovery from the filters is expected to be 100% based on analysis of organic standards, the OC/EC split into thermal–optical carbon analysis methods may introduce uncertainty into the TOR OC concentration (Chow et al., 2004). Approximate corrections for TOR/quartz sampling artifacts are made by various methods including using denuders and backup filters, and by subtracting blank OC concentrations, as in the IMPROVE and SEARCH networks (Chow et al., 2015). Residual OM, as discussed earlier,

encompasses substantial uncertainties due to various inputs such as particle water and nitrate sampling artifacts (Chow et al., 2015). Each method used in the present work to evaluate the FT-IR model results therefore has a mass recovery below 100% (as does the FT-IR calibration method), but approximates the total OM or OC.

The mass recoveries of OM and OC measured by FT-IR spectrometry were evaluated by comparison with residual OM and TOR OC, respectively. The OM mass recovery (versus residual OM) was 81±5% (±95% confidence interval), estimated as the

orthogonal least squares slope of the regression between the two OM estimates (Figure 3).



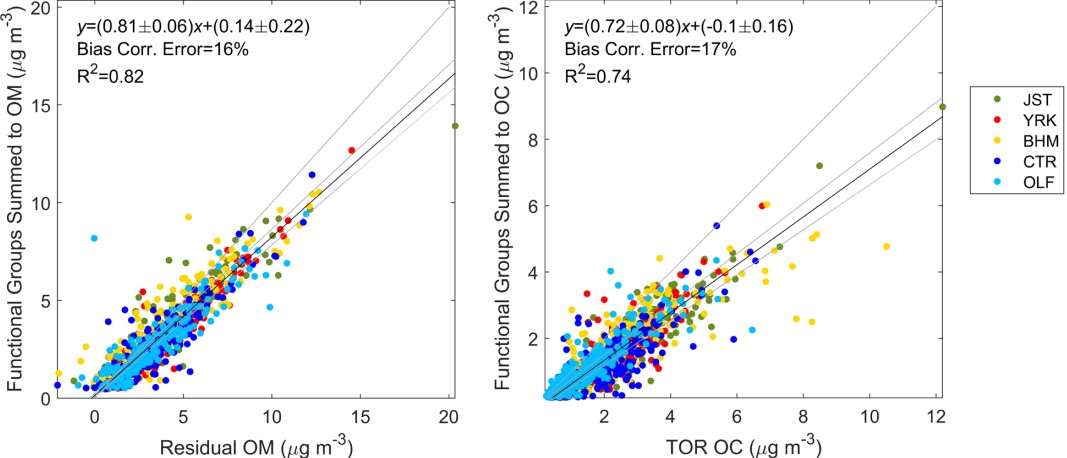

**Figure 3. Scatter plots of predicted functional group OM and OC concentrations versus reference measurement concentrations. Left: reference values are residual OM concentrations; FT-IR OM MDL=0.38 μg m⁻³. Right: reference values are TOR OC concentrations; FT-IR OC MDL=0.19 μg m⁻³. Site abbreviations: JST = Jefferson Street, Atlanta, GA; YRK = Yorkville, GA; BHM = Birmingham, AL; CTR = Centreville, AL; OLF = Outlying Landing Field (near Pensacola), FL.**

The correlation between the functional group and residual OM concentrations was strong ($R^2$=0.82) with a bias-corrected error of 16% (Figure 3), demonstrating that the modeled OM concentrations in the SEARCH ambient samples were consistent with, and accounted for most of, residual OM. Similarly, functional group OC accounted for 71±8% of TOR OC and was correlated with TOR OC concentrations ($R^2$=0.74), with a bias-corrected error of 17%.

### 3.4.2    Evaluating FT-IR measurements: OM and functional group concentrations

The median concentrations of OM estimated as the sum of functional groups in the SEARCH network were within the range of those measured previously using aerosol mass spectrometry and FT-IR spectrometry (~1–10 μg m⁻³; Kamruzzaman et al., 2018; Ruthenburg et al., 2014; Sun et al., 2011; Xu et al., 2015). Overall, OM contributed ~35% of PM$_{2.5}$ mass in the 2009–2016 SEARCH samples, ranging typically between 20–60% (interquartile range).  The median OM concentrations are greater at urban sites (JST, BHM) than at rural sites (CTR, YRK, OLF), as anticipated. The greater contribution of oxygenated functional groups (relative to previous FT-IR spectrometry aerosol composition studies) agreed with expectations that southeastern aerosol would be highly oxidized relative to OM from other parts of the country (e.g., Simon et al., 2011). Data from all analyzed SEARCH years will be further detailed in a forthcoming paper on trends in the dataset, but data from 2013 are discussed briefly here to demonstrate model improvements.

The 2013 SEARCH sample predictions were compared with previous model results from IMPROVE network sites in the southeastern US in 2013 to evaluate the similarity of the models (using work from Kamruzzaman et al., 2018 and Ruthenburg et al., 2014). The urban Alabama IMPROVE samples were collocated with the SEARCH BHM site, and other IMPROVE sites were located throughout the southeastern US. A comparison of urban and rural sites overall demonstrated that the current models predicted greater OM concentrations (3.4±3.0 μg m⁻³ at urban and 2.6±2.1 μg m⁻³ at rural sites; median ± interquartile range) than the 2014 models (2.1±2.0 μg m⁻³ urban and 0.74±0.67 μg m⁻³ rural). Median OM concentrations at Birmingham, AL were greater



using the current models (3.7±2.0 µg m⁻³) than the 2014 models (2.1±2.0 µg m⁻³). The greater predicted OM concentrations were attributed to the inclusion of a more extensive variety of organic molecules, and in particular to the greater variety of oxygenated functional groups in the present models (Figure 4).

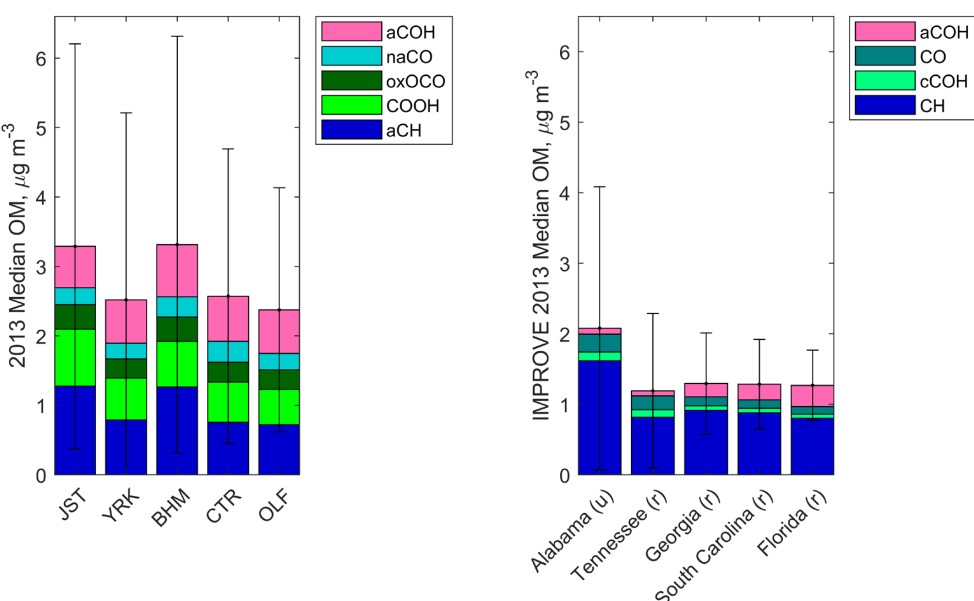

**Figure 4. Bar plots of OM median concentrations for all sites, 2013, using the current models and SEARCH ambient samples (left), as compared to previous work (applied by Kamruzzaman et al., 2018, to IMPROVE network samples, using models constructed by Ruthenburg et al., 2014). Error bars represent the interquartile ranges of the total OM concentrations.**

The concentrations of functional groups measured in ambient samples were in agreement with other atmospheric chemistry

observations. Functional group contributions to OM in SEARCH samples included ~25–45% aCH by mass. This large fraction was expected since nearly every molecule in organic aerosol contains aCH. The quantity of aCH attributed in the 2014 models was greater than in the current models, demonstrating the improvement to models of oxygenated functional groups. Carboxylic acids, followed by alcohols, also contributed substantially to the OM concentrations (~20–30% COOH and ~15–30% aCOH). It is unsurprising that these oxygenated functional groups were abundant in the samples, based on previous work in general

(Kawamura and Bikkina, 2016) and in the southeastern US (Gao et al., 2006). The median contribution of COOH to OM was lower at the urban sites than the rural sites, which can be attributed to the fresher emissions typically sampled at urban sites. Non-acid carbonyls also contributed 5–20%, and oxOCO contributed 5–10% of OM by mass; oxOCO concentrations were equivalent to ~40% of COOH concentrations (interquartile range 36–54%).  –

### 3.4.3    Evaluating FT-IR measurements: OM/OC ratios

The ratio of OM/OC by mass is a common metric for the degree of oxygenation of an ambient aerosol sample, and it is also used to estimate total OM concentrations from measured TOR OC concentrations (El-Zanan et al., 2009; Simon et al., 2011; Turpin



and Lim, 2001). The median OM/OC for all sites and years (*n*=1474 samples) was 2.1±0.2. As shown in Figure 5, the distributions of OM/OC ratios were slightly different between urban versus rural samples, and between winter (January) versus summer (July) samples. The urban versus rural differences may be muted because of a generally well-mixed atmosphere in the southeast (Gao et al., 2006; Weber et al., 2007; Xu et al., 2015). Seasonal OM/OC ratio differences in the southeast are also

5   likely small due to the narrow seasonal temperature variation in the southeastern US (Hidy et al., 2014). The greater values at rural sites (Yorkville 2.1±0.2 versus Atlanta 2.0±0.2, and Centreville 2.1±0.2 versus Birmingham 2.0±0.2) are in agreement with increased secondary organic aerosol contribution to OM downwind of urban emissions sources.

To assess the similarity of OM/OC ratios calculated from functional group concentrations with those of other methods, values were compared between the present results and other measurements from the southeast (Figure 5).

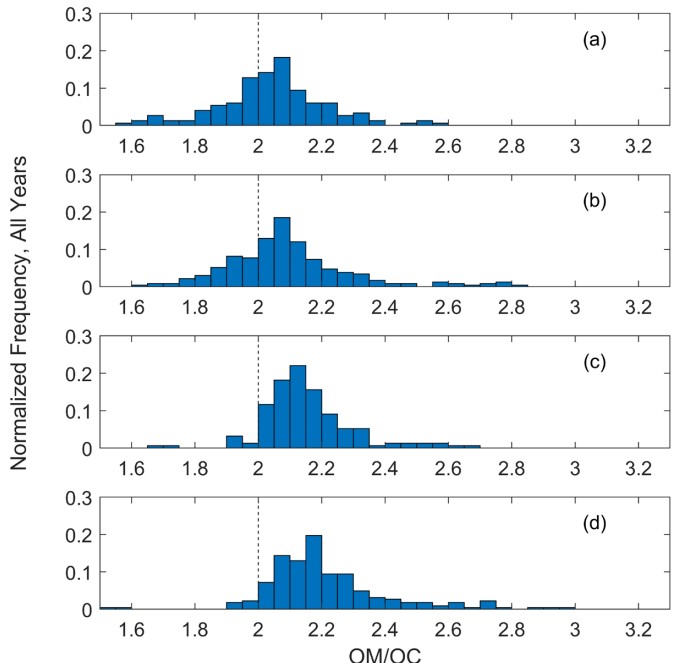

**Figure 5. Histograms of OM/OC ratios predicted using functional group measurements, separated into (a) urban (BHM and JST) January, (b) rural (CTR, YRK, and OLF) January, (c) urban July, and (d) rural July. January is used to approximate winter; July is used to approximate summer.**

15   The measured OM/OC ratios from the present models are similar to those estimated in another study using regression analysis: El-Zanan et al., 2009 measured OM/OC=2.16±0.43 (mean ± standard deviation) at JST between July 1998 and December 1999. However, the OM/OC values predicted using the current models were greater than those predicted using previous FT-IR models (Kamruzzaman et al., 2018): OM/OC=1.4±0.2 at urban Birmingham, AL and OM/OC=1.6±0.3 at four rural sites in the southeast (2013 IMPROVE sites). The increased OM/OC in the current work is attributable, again, to the added oxygenated chemicals

20   used to construct the current models, as well as the addition of the oxOCO functional group.





Extremes in measured OM/OC ratios were often caused by particularly high or low C-H stretching absorption intensity. Spectra of many high OM/OC ratio samples (>90$^{th}$ percentile of OM/OC) demonstrated low hydrocarbon character; these were mostly rural (~90%). Similarly, spectra of low OM/OC ratio samples (<10$^{th}$ percentile of OM/OC) demonstrated high hydrocarbon character and were ~90% urban. Additionally, some extreme values of OM/OC were characterized by low OM concentrations,

corresponding to functional group concentrations near or below MDLs.

### 3.4.4    Evaluating FT-IR measurements: O/C and H/C ratios

A van Krevelen diagram was generated using the atomic ratios of O/C and H/C from the functional groups quantified using FT-IR spectrometry (Figure 6). The range of mean O/C and H/C ratios measured by aerosol mass spectrometry in the southeastern US (Xu et al., 2015; summer 2013 data; designated by the black box in Figure 6), was near the visual mode of the FT-IR

predicted values. The organic composition observed using FT-IR spectrometry is thus chemically similar, as demonstrated in this space, to the aerosol captured by aerosol mass spectrometry. There are clearly methodological differences between FT-IR spectrometry and other mass spectrometry techniques used to characterize O/C and H/C ratios in other aerosol populations (summarized in Chen et al., 2015 and Heald et al., 2010). For example, the composition differs somewhat between PM$_{2.5}$ (analyzed by FT-IR spectrometry in this study) and PM$_1$ (as in aerosol mass spectrometry). However, sources are likely similar:

Liu et al. found similar sources of OM in PM$_1$ and PM$_{2.5}$ (2012). The agreement between results from the two methods in Figure 6 demonstrates that the overall chemical composition captured by the FT-IR spectrometry and mass spectrometry techniques is generally alike.

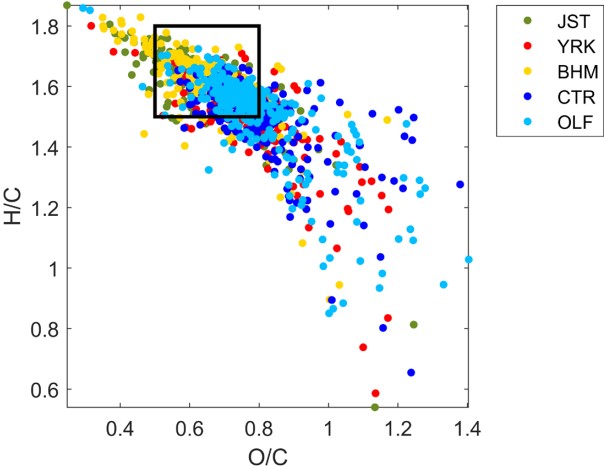

**Figure 6. Ratios of H/C and O/C measured in SEARCH ambient samples using FT-IR spectrometry models, plotted in the van**
**Krevelen space. The bold black box surrounds the data range collected using aerosol mass spectrometry in the southeastern US during the summer of 2013 (Xu et al., 2015).**



Aerosol evaluated in this study was less oxidized at the urban sites than the rural sites, as expected: the urban sites had higher H/C and lower O/C ratios (yellow and green in Figure 6) than the rural sites. A regional oxidized aerosol character was suggested by the similarity between the van Krevelen spaces occupied by the three rural sites. The spread in O/C and H/C values in Figure 6 was reflective of composition and was within the range observed in previous atmospheric aerosol studies (Chen et al., 2015);

some scatter was expected over the variety of seasons, sources and oxidation processes encompassed by this dataset. Extreme data points within the van Krevelen space (H/C≤1.2 and/or O/C≥1.2) generally corresponded to spectra with high Si concentrations (measured in the SEARCH network via X-ray fluorescence; see discussion in Supplementary Material, Sect. 14), low organic feature absorption (below median C-H and C=O stretching absorption) and/or a low OC concentration (FT-IR spectrometry or TOR). Although high Si concentrations apparently coincided with extreme O/C (high) and H/C (low) values,

these samples were kept in the dataset because there was no other indication of compromised prediction for these samples. Elevated (>95[th] percentile) Si concentrations were often observed during summer Saharan dust transport events (Hand et al., 2017), and narrow SiO-H stretching bands are observed in FT-IR spectra between 3600 and 3700 cm$^{-1}$.

The overall slope of all sites and years of SEARCH samples in the van Krevelen space is -0.72 (orthogonal least squares), which is similar to the slope measured over multiple, globally spaced field campaigns using aerosol mass spectrometry (-0.6; -1 to -0.7

in individual campaigns; Chen et al., 2015). The observed overall slope is intermediate between a van Krevelen space slope of -1 and -0.5, which could respectively approximate replacing a methyl group with a carboxylic acid group (the addition of two O and removal of two H atoms), and fragmentation of a C-C bond and formation of two carboxylic acid groups (Ng et al., 2011). The pattern of the current long-term dataset, which demonstrates atmospheric organic chemical composition integrated over many sources and atmospheric processes, is therefore consistent with common oxidation mechanisms observed in previous studies.

**3.5  Method limitations and future work**

The expansion of the list of chemicals included in the current calibration models from previous work was overall successful relative to the techniques and other study results discussed in the previous sections. These improvements suggest that further expanding and refining the calibration standards may make functional group measurements more accurate, increase the number of functional groups that can be measured, and overall improve OM recovery. Other chemicals suggested for addition to the

models include those used for aerosol mass spectrometry calibration (Aiken et al., 2008; Canagaratna et al., 2015). Additionally, several oxygenated chemicals with multiple functional groups proved difficult to collect in relevant chemical form and quantity for atmospheric aerosol (e.g., tartaric acid; see Supplementary Material, Sect. 4 and Fig. S-4), but should be revisited as atmospherically important species/groups. Other potentially important groups to consider include aldehydes and anhydrides (see Supplementary Material, Sect. 4). Since it was discovered herein that high silicate (dust) concentrations might degrade the

quality of functional group predictions, building calibrations of suspended dust particles could be considered in the future.
Although the extended calibration designed in this work captures variation in the carbon speciation beyond that of previous work, there are additional functional groups that should be considered in further work. Kamruzzaman et al. (2018) demonstrated the importance of amine N-H and C-N bonds in OM calculations, and organosulfate O-S and O-C bonds are additionally likely to be influential (Stone et al., 2012).

The observed interactions between some polar, protic species are acknowledged and should be considered in future work. However, the uncertainties in quantitative multi-chemical laboratory standards prevented us from including them in the current



models. The challenges observed in our explorations have included: (1) an inability to measure the weight of *each* chemical collected from particles generated from a solution with both chemicals; and (2) volatilization during sequential collection of chemicals. Multi-component laboratory standards could be a way to include atmospheric chemical interactions in FT-IR spectrometry models if the above challenges can be overcome or sufficiently minimized.

There are additional sources of uncertainty within the model parameters that should be further addressed. Takahama and Ruggeri (2017) demonstrated that the C/functional group ($\lambda$) values applied to calculate OM and OC concentrations herein are likely realistic, but have some uncertainty and are known to vary for different chemicals. Although the number of standards per chemical, the number of factors in each PLS model, the dynamic range of standards included in each model, and other inputs have been selected carefully, the correct values of these inputs cannot be exactly known. The chemical selection uncertainty

addressed in Sect. 3.3.2 cannot entirely capture the variability in model results due to the possible mis-specification of chemicals used in the models, and could be further discussed. Despite these challenges, the agreement between our results and many available expected or reference values, demonstrates that the results are reasonable (e.g., OM and OC concentrations with previous work, given mass accuracy expectations, realistic trends in urban versus rural concentrations, and atomic ratios O/C, H/C).

**4   Conclusions**

A method of directly estimating OM concentrations and OM/OC ratios with functional group composition has been advanced and evaluated. A multivariate calibration for quantifying five organic functional groups was built using FT-IR spectra and gravimetric weights of chemical standard filters. Spectra and weights of 18 organic chemicals and three interferent chemicals (ammonium sulfate, ammonium nitrate and particle water) were included in the calibration models. Various uncertainties in the

method were explored, such as humidity and hydrogen bonding differences between standard spectra. Ambient aerosol composition was quantified from nearly 1500 SEARCH network samples. An estimate of sampling uncertainty was calculated as precision between measurements from collocated sites (0.38 μg m$^{-3}$ or 14% of OM). The method gave results comparable to more intensive or sample destructive methods such as OM concentration via summation of various analytical results (residual OM), OC concentration via TOR, O/C ratio via aerosol mass spectrometry, and OM/OC via filter extraction and chromatography

analyses (functional group models accounted for 81±5% of residual OM, $R^2$=0.82, and 71±8% of TOR OC, $R^2$=0.74). Predictive features in the model excluded inorganic absorption features prominent in atmospheric aerosol FT-IR spectra (for example, bands due to ammonium sulfate and ammonium nitrate). Estimated functional group composition contained predominantly aliphatic C-H and carboxylic acid groups, followed by alcohol groups. Oxalates were quantified separately from carboxylic acids and contributed 5–10% of OM mass (~40% as much as carboxylic acids). Urban and rural SEARCH site compositions were

distinct, with a smaller aCH fraction, greater oxygenated functional group fraction, and lower OM concentration at rural sites. Further analysis of the SEARCH network data, including trends in OM concentration and composition observed between 2009 and 2016, will be explored in a forthcoming paper.

**Competing interests**

The authors declare that they have no conflict of interest.



**Acknowledgements**

Funding for this project was generously provided by the Electric Power Research Institute and with equipment and logistical support from Atmospheric Research & Analysis, Inc. The authors would like to acknowledge the contributions of Kelsey Seibert, who provided extensive lab management and data support for this work. Many undergraduate students were also involved in this

project, including Nathaniel Hopper, Matthew Coates, Alex Williams, and Kimberly Bowman.

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
