# Peer review of "Quantifying organic matter and functional groups in particulate matter filter samples from the southeastern United States, part I: Methods"

_Atmospheric Measurement Techniques, 2019_

## Referee Comment (RC1) · Anonymous Referee #1 · 20 May 2019

This paper reports a more comprehensive and more accurate analysis of the functional group content of aerosol samples measured by FTIR of samples on Teflon filters than presented to date. For this reason, it is an important advance and should be published.

There are a few minor issues that should be fixed (citation to unpublished work p. 5; discarding of outliers p.6; incorrect wording "compromise" p. 20) but nothing major. Overall, this is a tour de force of analytical chemistry with modern statistical methods applied, which result in new calibrations and improved fitting. The work as written is comprehensive, complete, and accurate.

My main and only quibble is that while the work is undoubtedly an improvement over

past work it fails to provide a direct comparison to other methods cited. So the reader is left to wonder the degree to which the new calibrations and improved fitting affect the results. I realize this is only a relative standard, but it still seems of merit with respect to existing literature. Does it change other results by 10% or more? Or does it provide a much more substantive analysis that is consistent with past findings?

---

## Referee Comment (RC2) · 20 May 2019

Based on a series of previous studies (such as Takahama), the author improved the current FTIR measurement method for quantifying organic aerosols. The research topic is of great significance. It can be used not only to quantify the mass concentration of organic compounds, but also quantify the chemical functional group information of organic aerosols. Here is a suggestion to improve this paper. It is hoped that the author can analyze and determine the functional groups in at least a few different samples by other instruments, such as NMR or HR-AMS. The results were used to confirm that the author's FTIR method for the determination of functional groups can be matched with

the non-conventional analysis methods.

---

## Author Comment (AC1) · 7 Aug 2019

Referee comment:

This paper reports a more comprehensive and more accurate analysis of the functional group content of aerosol samples measured by FTIR of samples on Teflon filters than presented to date. For this reason, it is an important advance and should be published.

Author response:

We appreciate your positive assessment of our work.

Referee comment:

[Figure]

There are a few minor issues that should be fixed (citation to unpublished work p. 5; discarding of outliers p.6; incorrect wording "compromise" p. 20) but nothing major.

Author response:

With respect to these particular minor issues:

- The unpublished work by Burki et al., on page 5 is unfortunately not yet in publication.

- Reference to the discussion of outlier handling was clarified on page 6.

- Changed original text from: "FT-IR spectra were acquired (Sect. 2.3), and outliers were detected; these were either set aside during model development or removed from the dataset (Sect. 2.4)." The following revised sentence clarifies the summary of outlier handling: "After FT-IR spectra were acquired (Sect. 2.3), outliers were detected and were either set aside during model development or removed from the dataset (Sect. 2.4)."

- The word "compromise" has been corrected to "comprise" on page 20.

Referee comment:

Overall, this is a tour de force of analytical chemistry with modern statistical methods applied, which result in new calibrations and improved fitting. The work as written is comprehensive, complete, and accurate.

Author response:

Thank you very much for these supportive comments.

Referee comment:

My main and only quibble is that while the work is undoubtedly an improvement over past work it fails to provide a direct comparison to other methods cited. So the reader is left to wonder the degree to which the new calibrations and improved fitting affect the results. I realize this is only a relative standard, but it still seems of merit with respect

to existing literature. Does it change other results by 10% or more? Or does it provide a much more substantive analysis that is consistent with past findings?

Author response:

We recognize that there are no directly comparable functional group concentrations quantified via other methods. However, for this work, we instead have focused on understanding the origins of model sensitivity to composition of calibration standards, and evaluated the outcome of our decisions against TOR OC and residual OM concentrations based on their availability for the same sites and dates as our FT-IR measurements (∼1000 samples; Section 3.4.1). Measurements from the southeastern US were compared to our study results in van Krevelen Space (Section 3.4.4). We have additionally compared the results of our new models to previous OM and functional group model results from our group using collocated IMPROVE and SEARCH samplers at the Birmingham, Alabama site. This is a direct comparison of our early FT-IR functional group models to the one presented here (Section 3.4.2). Additional comparisons between IMPROVE sites in the southeastern US and SEARCH sites are made (Section 3.4.2). Previous FT-IR functional group methods developed by our group are not suitable for application to the SEARCH filters due to a difference in Teflon filter thickness, which greatly diminishes the FT-IR absorbance spectral intensity.

A "shootout"-type comparison against several other methods for a more restricted measurement campaign can be envisioned in the future, and will build upon the work presented here in which FT-IR model building practices are further developed. In the meantime, the following points discuss material we have added or revised within the paper to highlight direct comparisons to other available measurements and previous FT-IR models.

1. We have added a clause in our abstract highlighting the comparison of our results with those of other works: "We have built FT-IR spectrometry functional group calibration models that improve upon previous work, as demonstrated by the comparison of

current model results with those of previous models and other OM analysis methods."

2. We have changed the title of Section 3.4 from: "Using aerosol composition to evaluate FT-IR functional group measurements" to: "Evaluation by comparison to other methods and previous FT-IR spectrometry work".

3. In our comparison to previous work using FT-IR spectrometry models (work by Ruthenburg et al., 2014 and Kamruzzaman et al., 2018), we have focused foremost on the Birmingham site since it is collocated with an IMPROVE site; the latter is reported on by Kamruzzaman et al., 2018 (Section 3.4.2, pages 28-29):

"The functional group composition of OM at the Birmingham, Alabama IMPROVE site measured using the previous FT-IR spectrometry models (Figure 4, right panel) were compared to the collocated SEARCH BHM samples (concentrations measured using the current models). Median OM concentrations at Birmingham were greater using the current models ($3.1\pm2.8$ $\mu$g mˆ-3) than the 2014 models ($2.1\pm2.0$ $\mu$g mˆ-3), by 48%. The greater OM concentrations predicted by the current models can be explained mainly by enhanced oxygenated functional group concentrations: while the contributions of aCH to OM concentrations were lower at Birmingham using the current model predictions (median concentrations were 1.20 $\mu$g mˆ-3 versus 1.62 $\mu$g mˆ-3 in 2013 current and previous models, respectively), the oxygenated functional groups are all substantially higher (1.91 $\mu$g mˆ-3 versus 0.46 $\mu$g mˆ-3 respectively). In particular, oxOCO accounted for $\sim$10% of OM in the current models (0.32 $\mu$g mˆ-3), adding substantially to the quantified material."

4. In Section 3.4.1, we have added a concluding sentence citing other work that supports the calculated mass recoveries: "The mass recovery observed in this work is similar to that in previous FT-IR spectrometry measurements (Takahama and Ruggeri, 2017)."

5. We have added a summary of findings from the 2013 SOAS campaign comparing AMS and FT-IR spectrometry measurements (Section 3.4.1):

"A study comparing simultaneous characterization of OM composition found that the FT-IR spectrometry OM concentrations were 20-40% lower than those observed using aerosol mass spectrometry, within the combined uncertainties of the methods ($\sim$20% for each method; Liu et al., 2018)."

6. Finally, we have included additional literature on OM/OC ratio measurements for comparison to those observed in the present work. The paragraph is now as follows (Section 3.4.3, pages 30-31):

"The measured OM/OC ratios from the present models are similar to those estimated in another study: El-Zanan et al., 2009 measured OM/OC=2.16$\pm$0.43 and OM/OC=2.14$\pm$017 at JST between July 1998 and December 1999 (mean $\pm$ standard deviation; using gravimetric analysis of solvent extracts and mass balance of organic and total particulate masses, respectively). Multiple linear regression has been applied to IMPROVE data to obtain OM/OC at various locations and times and has resulted in varying values. Simon et al., 2011 found lower median seasonal OM/OC ratios for the southeastern US (between 1.64 and 1.89). An OM/OC of 1.8, used to calculate reconstructed fine mass concentrations within IMPROVE network samples (Pitchford et al., 2007), is also lower than the median OM/OC ratios estimated in this study. However, in more recent work using multiple linear regression, Hand et al. (2019) estimated OM/OC ratios in the southeastern US varying between 1.9 and 2.1 from 2012 to 2016, similar to the ratios presented in this present work. The OM/OC values predicted using previous FT-IR models (Kamruzzaman et al., 2018) were lower than those of the other approaches summarized here and the current FT-IR model results: OM/OC=1.4$\pm$0.2 at urban Birmingham, AL and OM/OC=1.6$\pm$0.3 at four rural sites in the southeast (2013 IMPROVE sites). The higher OM/OC ratios in the current work are attributable to the added oxygenated chemicals used to construct the current models and the addition of the oxOCO functional group."

References:

El-Zanan, H. S., Zielinska, B., Mazzoleni, L. R. and Hansen, D. A.: Analytical determination of the aerosol organic mass-to-organic carbon ratio, J. Air Waste Manag. Assoc., 59(1), 58–69, doi:10.3155/1047-3289.59.1.58, 2009.

Hand, J. L., Prenni, A. J., Schichtel, B. A., Malm, W. C. and Chow, J. C.: Trends in remote PM 2.5 residual mass across the United States: Implications for aerosol mass reconstruction in the IMPROVE network, Atmos. Environ., 203(January), 141–152, doi:10.1016/j.atmosenv.2019.01.049, 2019.

Kamruzzaman, M., Takahama, S. and Dillner, A. M.: Quantification of amine functional groups and their influence on OM/OC in the IMPROVE network, Atmos. Environ., 172, 124–132, doi:10.1016/j.atmosenv.2017.10.053, 2018.

Liu, J., Russell, L. M., Ruggeri, G., Takahama, S., Claflin, M. S., Ziemann, P. J., Pye, H. O. T., Murphy, B. N., Xu, L., Ng, N. L., McKinney, K. A., Budisulistiorini, S. H., Bertram, T. H., Nenes, A. and Surratt, J. D.: Regional Similarities and NOx-Related Increases in Biogenic Secondary Organic Aerosol in Summertime Southeastern United States, J. Geophys. Res. Atmos., 123(18), 10,620-10,636, doi:10.1029/2018JD028491, 2018.

Pitchford, M., Malm, W., Schichtel, B., Kumar, N., Lowenthal, D. and Hand, J.: Revised algorithm for estimating light extinction from IMPROVE particle speciation data, J. Air Waste Manag. Assoc., 57(11), 1326–1336, doi:10.3155/1047-3289.57.11.1326, 2007.

Ruthenburg, T. C., Perlin, P. C., Liu, V., McDade, C. E. and Dillner, A. M.: Determination of organic matter and organic matter to organic carbon ratios by infrared spectroscopy with application to selected sites in the IMPROVE network, Atmos. Environ., 86, 47–57, doi:10.1016/j.atmosenv.2013.12.034, 2014.

Simon, H., Bhave, P. V, Swall, J. L., Frank, N. H. and Malm, W. C.: Determining the spatial and seasonal variability in OM/OC ratios across the US using multiple regression, Atmos. Chem. Phys., 11, 2933–2949, doi:10.5194/acp-11-2933-2011, 2011.

Takahama, S. and Ruggeri, G.: Technical note: Relating functional group measurements to carbon types for improved model-measurement comparisons of organic aerosol composition, Atmos. Chem. Phys., 17(7), 4433–4450, doi:10.5194/acp-17-4433-2017, 2017.

---

## Author Comment (AC2) · 7 Aug 2019

Referee comment:

Based on a series of previous studies (such as Takahama), the author improved the current FTIR measurement method for quantifying organic aerosols. The research topic is of great significance. It can be used not only to quantify the mass concentration of organic compounds, but also quantify the chemical functional group information of organic aerosols.

Author response:

Thank you for your kind summary of, and comments regarding, our work.

Referee comment:

Here is a suggestion to improve this paper. It is hoped that the author can analyze and determine the functional groups in at least a few different samples by other instruments, such as NMR or HR-AMS. The results were used to confirm that the author's FTIR method for the determination of functional groups can be matched with the non-conventional analysis methods.

Author response:

We appreciate this insightful suggestion. Although a comparison to other direct measurements of OM in the same sampling time periods and locations such as NMR or AMS was not possible for this work, we have compared to various southeastern US aerosol measurements: AMS measurements of O/C and H/C ratios (using the van Krevelen space; Section 3.4.4), various measurements of OM/OC, as well as residual OM and TOR OC measurements (Sections 3.4.1 and 3.4.3). In addition to these comparisons, our introduction mentions the work of previous studies directly comparing AMS fragment and FT-IR spectrometry functional group abundances. We have responded to a similar comment made by Reviewer #1, and respectfully request that Reviewer #2 also view our response to Reviewer #1.

References

El-Zanan, H. S., Zielinska, B., Mazzoleni, L. R. and Hansen, D. A.: Analytical determination of the aerosol organic mass-to-organic carbon ratio, J. Air Waste Manag. Assoc., 59(1), 58–69, doi:10.3155/1047-3289.59.1.58, 2009.

Hand, J. L., Prenni, A. J., Schichtel, B. A., Malm, W. C. and Chow, J. C.: Trends in remote PM 2.5 residual mass across the United States: Implications for aerosol mass reconstruction in the IMPROVE network, Atmos. Environ., 203(January), 141–152, doi:10.1016/j.atmosenv.2019.01.049, 2019.
Kamruzzaman, M., Takahama, S. and Dillner, A. M.: Quantification of amine functional groups and their influence on OM/OC in the IMPROVE network, Atmos. Environ., 172, 124–132, doi:10.1016/j.atmosenv.2017.10.053, 2018.

Liu, J., Russell, L. M., Ruggeri, G., Takahama, S., Claflin, M. S., Ziemann, P. J., Pye, H. O. T., Murphy, B. N., Xu, L., Ng, N. L., McKinney, K. A., Budisulistiorini, S. H., Bertram, T. H., Nenes, A. and Surratt, J. D.: Regional Similarities and NOx-Related Increases in Biogenic Secondary Organic Aerosol in Summertime Southeastern United States, J. Geophys. Res. Atmos., 123(18), 10,620-10,636, doi:10.1029/2018JD028491, 2018.

Pitchford, M., Malm, W., Schichtel, B., Kumar, N., Lowenthal, D. and Hand, J.: Revised algorithm for estimating light extinction from IMPROVE particle speciation data, J. Air Waste Manag. Assoc., 57(11), 1326–1336, doi:10.3155/1047-3289.57.11.1326, 2007.

Ruthenburg, T. C., Perlin, P. C., Liu, V., McDade, C. E. and Dillner, A. M.: Determination of organic matter and organic matter to organic carbon ratios by infrared spectroscopy with application to selected sites in the IMPROVE network, Atmos. Environ., 86, 47–57, doi:10.1016/j.atmosenv.2013.12.034, 2014.

Simon, H., Bhave, P. V, Swall, J. L., Frank, N. H. and Malm, W. C.: Determining the spatial and seasonal variability in OM/OC ratios across the US using multiple regression, Atmos. Chem. Phys., 11, 2933–2949, doi:10.5194/acp-11-2933-2011, 2011.

Takahama, S. and Ruggeri, G.: Technical note: Relating functional group measurements to carbon types for improved model-measurement comparisons of organic aerosol composition, Atmos. Chem. Phys., 17(7), 4433–4450, doi:10.5194/acp-17-4433-2017, 2017.

---

## Author Response (AR1)

**Response to Reviewers: for "Quantifying organic matter and functional groups in particulate matter filter samples from the southeastern United States, part I: Methods" by A. J. Boris et al.**

**Interactive comment from Anonymous Referee #1**

This paper reports a more comprehensive and more accurate analysis of the functional group content of aerosol samples measured by FTIR of samples on Teflon filters than presented to date. For this reason, it is an important advance and should be published.

*We appreciate your positive assessment of our work.*

There are a few minor issues that should be fixed (citation to unpublished work p. 5; discarding of outliers p.6; incorrect wording "compromise" p. 20) but nothing major.

*With respect to these particular minor issues:*
- *The unpublished work by Burki et al., on page 5 is unfortunately not yet in publication.*
- *Reference to the discussion of outlier handling was clarified on page 6.*
    - *Changed original text from:* "FT-IR spectra were acquired (Sect. 2.3), and outliers were detected; these were either set aside during model development or removed from the dataset (Sect. 2.4)." *The following revised sentence clarifies the summary of outlier handling:* "After FT-IR spectra were acquired (Sect. 2.3), outliers were detected and were either set aside during model development or removed from the dataset (Sect. 2.4)."
- *The word "compromise" has been corrected to "comprise" on page 20.*

Overall, this is a tour de force of analytical chemistry with modern statistical methods applied, which result in new calibrations and improved fitting. The work as written is comprehensive, complete, and accurate.

*Thank you very much for these supportive comments.*

My main and only quibble is that while the work is undoubtedly an improvement over past work it fails to provide a direct comparison to other methods cited. So the reader is left to wonder the degree to which the new calibrations and improved fitting affect the results. I realize this is only a relative standard, but it still seems of merit with respect to existing literature. Does it change other results by 10% or more? Or does it provide a much more substantive analysis that is consistent with past findings?

*We recognize that there are no directly comparable functional group concentrations quantified via other methods. However, for this work, we instead have focused on understanding the origins of model sensitivity to composition of calibration standards, and evaluated the outcome of our decisions against TOR OC and residual OM concentrations based on their availability for the same sites and dates as our FT-IR measurements (~1000 samples; Section 3.4.1). Measurements from the southeastern US were compared to our study results in van Krevelen Space (Section 3.4.4). We have additionally compared the results of our new models to previous OM and functional group model results from our group using collocated IMPROVE and SEARCH samplers at the Birmingham, Alabama site. This is a direct comparison of our early FT-IR functional group models to the one presented here (Section 3.4.2). Additional comparisons between IMPROVE sites in the southeastern US and SEARCH sites are made (Section 3.4.2).*

*Previous FT-IR functional group methods developed by our group are not suitable for application to the SEARCH filters due to a difference in Teflon filter thickness, which greatly diminishes the FT-IR absorbance spectral intensity.*

*A "shootout"-type comparison against several other methods for a more restricted measurement campaign can be envisioned in the future, and will build upon the work presented here in which FT-IR model building practices are further developed. In the meantime, the following points discuss material we have added or revised within the paper to highlight direct comparisons to other available measurements and previous FT-IR models.*

1. *We have added a clause in our abstract highlighting the comparison of our results with those of other works: "We have built FT-IR spectrometry functional group calibration models that improve upon previous work, as demonstrated by the comparison of current model results with those of previous models and other OM analysis methods."*

2. *We have changed the title of Section 3.4 from: "Using aerosol composition to evaluate FT-IR functional group measurements" to: "Evaluation by comparison to other methods and previous FT-IR spectrometry work".*

3. *In our comparison to previous work using FT-IR spectrometry models (work by Ruthenburg et al., 2014 and Kamruzzaman et al., 2018), we have focused foremost on the Birmingham site since it is collocated with an IMPROVE site; the latter is reported on by Kamruzzaman et al., 2018 (Section 3.4.2, pages 28-29):*

   "The functional group composition of OM at the Birmingham, Alabama IMPROVE site measured using the previous FT-IR spectrometry models (Figure 4, right panel) were compared to the collocated SEARCH BHM samples (concentrations measured using the current models). Median OM concentrations at Birmingham were greater using the current models ($3.1\pm2.8$ μg m$^{-3}$) than the 2014 models ($2.1\pm2.0$ μg m$^{-3}$), by 48%. The greater OM concentrations predicted by the current models can be explained mainly by enhanced oxygenated functional group concentrations: while the contributions of aCH to OM concentrations were lower at Birmingham using the current model predictions (median concentrations were 1.20 μg m$^{-3}$ versus 1.62 μg m$^{-3}$ in 2013 current and previous models, respectively), the oxygenated functional groups are all substantially higher (1.91 μg m$^{-3}$ versus 0.46 μg m$^{-3}$ respectively). In particular, oxOCO accounted for ~10% of OM in the current models (0.32 μg m$^{-3}$), adding substantially to the quantified material."

4. *In Section 3.4.1, we have added a concluding sentence citing other work that supports the calculated mass recoveries: "The mass recovery observed in this work is similar to that in previous FT-IR spectrometry measurements (Takahama and Ruggeri, 2017)."*

5. *We have added a summary of findings from the 2013 SOAS campaign comparing AMS and FT-IR spectrometry measurements (Section 3.4.1):*

   "A study comparing simultaneous characterization of OM composition found that the FT-IR spectrometry OM concentrations were 20-40% lower than those observed using aerosol mass spectrometry, within the combined uncertainties of the methods (~20% for each method; Liu et al., 2018)."

6. *Finally, we have included additional literature on OM/OC ratio measurements for comparison to those observed in the present work. The paragraph is now as follows (Section 3.4.3, pages 30-31):*

"The measured OM/OC ratios from the present models are similar to those estimated in another study: El-Zanan et al., 2009 measured OM/OC=2.16±0.43 and OM/OC=2.14±017 at JST between July 1998 and December 1999 (mean ± standard deviation; using gravimetric analysis of solvent extracts and mass balance of organic and total particulate masses, respectively). Multiple linear regression has been applied to IMPROVE data to obtain OM/OC at various locations and times and has resulted in varying values. Simon et al., 2011 found lower median seasonal OM/OC ratios for the southeastern US (between 1.64 and 1.89). An OM/OC of 1.8, used to calculate reconstructed fine mass concentrations within IMPROVE network samples (Pitchford et al., 2007), is also lower than the median OM/OC ratios estimated in this study. However, in more recent work using multiple linear regression, Hand et al. (2019) estimated OM/OC ratios in the southeastern US varying between 1.9 and 2.1 from 2012 to 2016, similar to the ratios presented in this present work. The OM/OC values predicted using previous FT-IR models (Kamruzzaman et al., 2018) were lower than those of the other approaches summarized here and the current FT-IR model results: OM/OC=1.4±0.2 at urban Birmingham, AL and OM/OC=1.6±0.3 at four rural sites in the southeast (2013 IMPROVE sites). The higher OM/OC ratios in the current work are attributable to the added oxygenated chemicals used to construct the current models and the addition of the oxOCO functional group."

**Interactive comment from Qingcai Chen (Referee) chenqingcai@sust.edu.cn**

Based on a series of previous studies (such as Takahama), the author improved the current FTIR measurement method for quantifying organic aerosols. The research topic is of great significance. It can be used not only to quantify the mass concentration of organic compounds, but also quantify the chemical functional group information of organic aerosols.

*Thank you for your kind summary of, and comments regarding, our work.*

Here is a suggestion to improve this paper. It is hoped that the author can analyze and determine the functional groups in at least a few different samples by other instruments, such as NMR or HR-AMS. The results were used to confirm that the author's FTIR method for the determination of functional groups can be matched with the non-conventional analysis methods.

*We appreciate this insightful suggestion. Although a comparison to other direct measurements of OM in the same sampling time periods and locations such as NMR or AMS was not possible for this work, we have compared to various southeastern US aerosol measurements: AMS measurements of O/C and H/C ratios (using the van Krevelen space; Section 3.4.4), various measurements of OM/OC, as well as residual OM and TOR OC measurements (Sections 3.4.1 and 3.4.3). In addition to these comparisons, our introduction mentions the work of previous studies directly comparing AMS fragment and FT-IR spectrometry functional group abundances. We have responded to a similar comment made by Reviewer #1, and respectfully request that Reviewer #2 also view our response to Reviewer #1.*

**References**

[revised manuscript text omitted]
 ($3.4\pm3.0$ µg m$^{-3}$ at urban and $2.6\pm2.1$ µg m$^{-3}$ at rural sites; median $\pm$ interquartile range) than the 2014 models ($2.1\pm2.0$ µg m$^{-3}$ urban and $0.74\pm0.67$ µg m$^{-3}$ rural). Median OM concentrations at Birmingham, AL were greater using the current models ($3.7\pm2.0$ µg m$^{-3}$) than the 2014 models ($2.1\pm2.0$ µg m$^{-3}$). The greater predicted OM concentrations were attributed to the inclusion of a more extensive variety of organic molecules, and in particular to the greater variety of oxygenated functional groups in the present models (Figure 4).